# Role of Tim4 in the regulation of ABCA1⁺ adipose tissue macrophages and post-prandial cholesterol levels

M. S. Magalhaes[1], P. Smith [1], J. R. Portman[2], L. H. Jackson-Jones[1,3], C. C. Bain [2], P. Ramachandran [2], Z. Michailidou[1], R. H. Stimson [1], M. R. Dweck[1], L. Denby[1], N. C. Henderson [2,4], S. J. Jenkins [2] & C. Bénézech [1✉]

Dyslipidemia is a main driver of cardiovascular diseases. The ability of macrophages to scavenge excess lipids implicate them as mediators in this process and understanding the mechanisms underlying macrophage lipid metabolism is key to the development of new treatments. Here, we investigated how adipose tissue macrophages regulate post-prandial cholesterol transport. Single-cell RNA sequencing and protected bone marrow chimeras demonstrated that ingestion of lipids led to specific transcriptional activation of a population of resident macrophages expressing Lyve1, Tim4, and ABCA1. Blocking the phosphatidylserine receptor Tim4 inhibited lysosomal activation and the release of post-prandial high density lipoprotein cholesterol following a high fat meal. Both effects were recapitulated by chloroquine, an inhibitor of lysosomal function. Moreover, clodronate-mediated cell-depletion implicated Tim4⁺ resident adipose tissue macrophages in this process. Thus, these data indicate that Tim4 is a key regulator of post-prandial cholesterol transport and adipose tissue macrophage function and may represent a novel pathway to treat dyslipidemia.

[1] Centre for Cardiovascular Science, University of Edinburgh, Edinburgh, UK. [2] Centre for Inflammation Research, University of Edinburgh, Edinburgh, UK. [3] Division of Biomedical and Life Sciences, Lancaster University, Lancaster, UK. [4] MRC Human Genetics Unit, Institute of Genetics and Molecular Medicine, University of Edinburgh, Edinburgh, UK. ✉email: cbenezec@ed.ac.uk

The main function of adipose tissue (AT) is the storage of lipids to establish an energy reserve; adipocytes specialize in the uptake of dietary lipids and their storage as triglycerides (TGs). In the context of a diet mostly composed of low-calorie food, a meal particularly rich in lipid ("cheat" meal) is therefore a physiologic opportunity for adipocytes to increase their TG storage. While the role of adipose tissue macrophages (ATMs) in the metabolic adaptation to obesity is increasingly understood[1–4], little is known of the role of ATMs in the regulation of lipid metabolism and fat storage after a lipid-rich meal.

Efficient processing of a fat-containing meal is achieved through digestion and absorption of lipid nutrients in the gut and secretion of the lipid-transporting particles, chylomicrons, in the lymph. Chylomicrons are then delivered into the circulation via the thoracic duct, avoiding the portal circulation and facilitating their delivery to AT and muscle. The lipoprotein lipase (LPL), which is expressed at high levels in AT, hydrolyzes chylomicrons into fatty acids (FAs), allowing their preferential uptake and storage as TG in AT. This process generates chylomicron remnants, poor in TG and rich in cholesterol, which are highly atherogenic[5]. Macrophages readily accumulate lipid and cholesterol, a phenomenon driving fatty streak formation and evolution to atherosclerotic plaques in the vessel wall[6]. In the reverse cholesterol transport pathway, ABCA1 mediates the efflux of cholesterol and phospholipids to lipid-poor apolipoproteins (ApoA1 and ApoE), forming nascent high-density lipoproteins (HDLs), which facilitate the excretion of cholesterol[7]. Elevated circulating levels of chylomicron remnants and low-density lipoproteins (LDLs) are important risk factors for cardiovascular disease, while elevated levels of HDL cholesterol (HDLc) and efficient reverse cholesterol transport are protective[7–9]. ABCA1 is required for lipogenesis and lipid accretion in adipocytes during diet-induced obesity[10]. In hematopoietic cells, ABCA1 limits inflammation, the recruitment of monocytes and macrophages to AT, and protects against diet-induced insulin resistance[11]. In humans, obesity and insulin resistance have been associated with lower ABCA1 expression in AT[12].

Increased recruitment and retention of macrophages as well as in situ proliferation of ATMs contribute to accumulation of macrophages during prolonged high-fat diet (HFD), often with deleterious function in mouse and humans[13–16]. For instance, CD11c+ ATMs are associated with AT inflammation and insulin resistance[17–21]. However, recruitment of macrophages with high lysosomal-dependent lipid metabolism has a beneficial role in obese AT. Uptake and metabolism of excess lipid via lysosomal lipolysis in recruited Trem2+CD9+ ATMs, also called lipid-associated macrophages (LAMs), prevents adipocyte hypertrophy and adverse inflammation leading to metabolic dysregulation during obesity[22–24]. In lean mice, the AT is populated by a subset of resident Tim4+ ATMs closely associated with the vasculature, which has very high endocytic capacity, but whose function is not clear[25].

Genome-wide association studies (GWAS) have identified genetic variants of TIMD4 (T cell immunoglobulin mucin protein 4) associated with dyslipidemia. Tim4, a phosphatidylserine receptor, is present on numerous tissue-resident macrophages including the AT, but the relationship between dyslipidemia and Tim4 has not been elucidated[26–31].

Here we set out to investigate the effect of a lipid-rich meal on ATMs and to evaluate their function in the regulation of post-prandial lipid circulation. Using single-cell RNA sequencing (scRNA-seq) and protected bone marrow (BM) chimeras, we have demonstrated that, in lean mice, the ATM compartment was comprised of a number of transcriptionally distinct populations with varying dependence on blood monocytes for their replenishment. We confirmed that ATM residency was associated not only with increased endocytic capacity but also with increased lysosomal function and Abca1 expression. Ingestion of lipids led to transcriptional activation and increased lysosomal content of resident Lyve1+Tim4+ ATMs. Blocking Tim4 with anti-Tim4 immunoglobulin (Ig) inhibited the release of post-prandial HDLc and abrogated lysosomal activation in Lyve1+Tim4+ ATM. Both effects were recapitulated by chloroquine, an inhibitor of lysosomal function. Depletion of Tim4+ peritoneal macrophages and Tim4+ liver Kupffer cells using clodronate liposomes, which only partially depleted Tim4+ ATMs, did not affect post-prandial HDLc levels, indicating that peritoneal macrophages and Kupffer cells were not required to modulate HDLc levels and that Tim4+ macrophages from other tissues such as the AT were involved. The targeting of Tim4+ ATM metabolism may represent a novel therapeutic pathway to treat dyslipidemia and reduce the risk of atherosclerosis in humans.

## Results

**ScRNA-seq analysis reveals high heterogeneity of ATMs in lean mice.** To investigate the direct effect of lipid ingestion on ATMs, we performed unbiased scRNA-seq of ATMs harvested from the epididymal AT of mice fed overnight with a HFD and mice kept on control chow diet (CD). To maximize the transcriptional resolution of our analysis, we performed droplet-based scRNA-seq on isolated CD45+Lin−Ly6Clow/−F4/80+ macrophages (Fig. 1a). Unsupervised clustering based on shared and unique patterns of gene expression of 4358 ATMs from 6 fat pads (n = 3 CD pooled into 1 sample and n = 3 HFD pooled into 1 sample) identified 8 distinct populations that we visualized using uniform manifold approximation and projection (UMAP), revealing high ATM heterogeneity in lean mice (Fig. 1b). Each cluster contained cells from CD and HFD mice. Most ATMs clustered in 4 main populations (Clusters 1–4) (Fig. 1c). Cluster 1 showed low expression of Ccr2 and was distinguished by differentially expressed genes (DEGs), including Lyve1, Fcna, Folr2, Selenop, F13a1, Gas6, and Csf1r. This signature corresponded to a population of tissue-resident macrophages described in the AT, lungs, and heart[22,25,28,32]. Cluster 1 also showed expression of Timd4, albeit at low levels (Figs. 1d, e and S1a and Data file S1). Cluster 3–5 were distinguished by the expression of Ccr2, Lyz1, Ear2, and Retnla suggesting that these ATMs may represent cells recently derived from Ly6high monocytes. Compared with Clusters 4 and 5, Cluster 3 showed gradual increased expression of Adgre1, Lyve1, and Folr2 and progressive diminished expression of Ccr2, Lyz1, Ear2, and Retnla. Similarly, Cluster 2 appeared to be transcriptionally similar to Cluster 1 but with certain features of Cluster 3. Cluster 3 had relatively higher expression of antigen-presentation genes, such as H2-Eb1 and Cd74 (Figs. 1d, e and S1a and Data file S1). The graded pattern of the expression of genes such as Folr2, Lyz1, Ear2, and Nr4a1 across Clusters 1–5 suggested that these ATMs may be developmentally related (Fig. 1e). Cluster 5 was distinguished by high expression of Nr4a1, which increases transiently during the differentiation of Ly6Chigh monocytes into Ly6ClowF4/80+ macrophages[33,34] suggesting that ATMs from Cluster 5 were the most recently derived from monocytes.

We explored this hypothetical developmental relationship by performing lineage inference with slingshot, using Cluster 5 (Nr4a1high ATMs) as the starting cluster. Projection of pseudotime on the UMAP plot confirmed that ATMs followed a pseudotime trajectory straddling Cluster 5–1 in mice kept on CD (Fig. 1f, g). To track changes across this trajectory, gene expression was plotted as a function of pseudotime. This analysis showed the gradual downregulation of genes highly expressed by Cluster 5 such as Fn1, Ear2, and Lyz1 and gradual increased

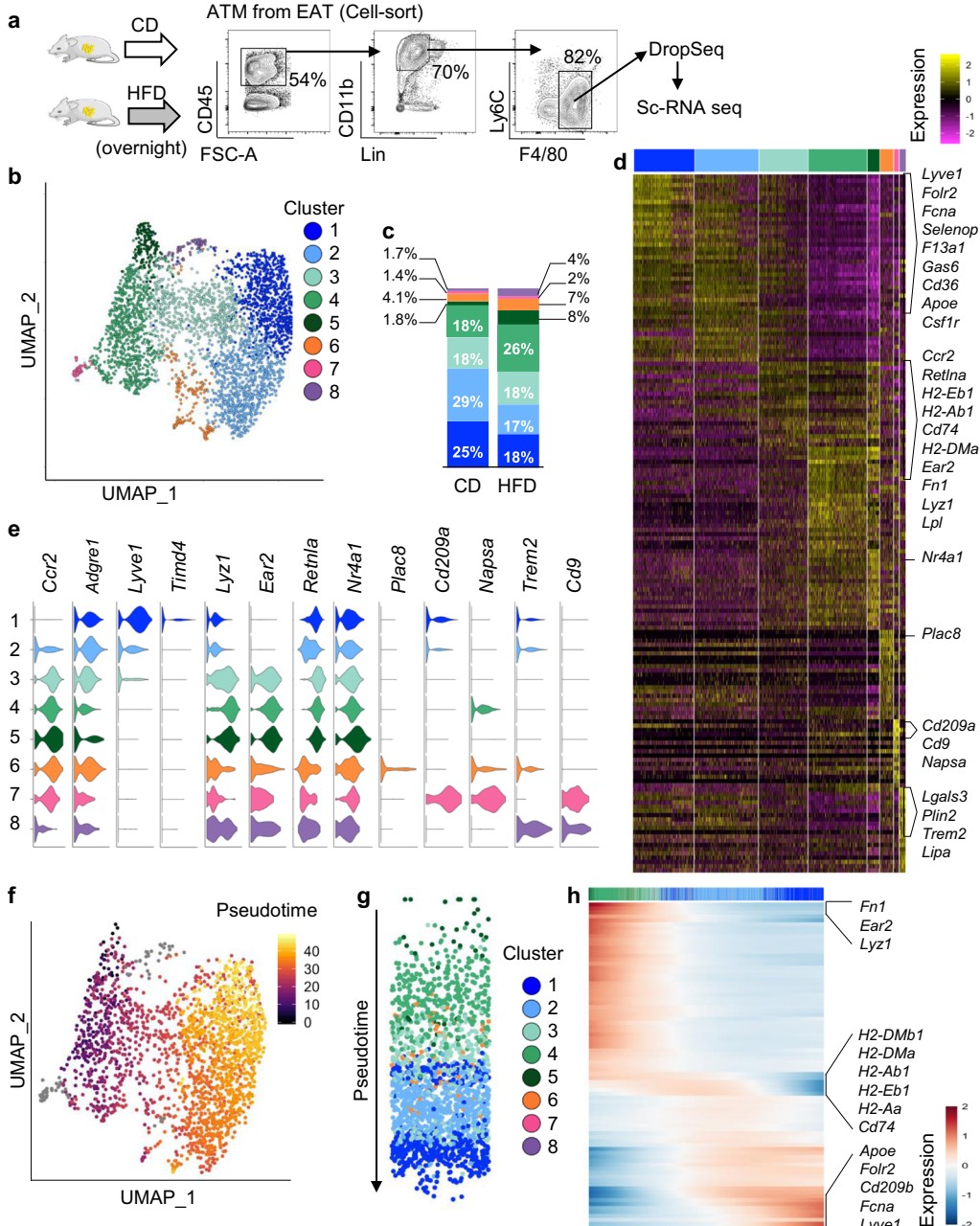

**Fig. 1 ScRNA-seq characterization of ATMs. a** CD45$^+$Lin$^-$CD11b$^+$F4/80$^+$ ATMs from the epididymal AT (EAT) of mice kept on CD ($n = 3$) or fed a HFD overnight ($n = 3$) were cell-sorted and underwent scRNA-seq. Lineage (Lin) includes TCRb, CD19, SiglecF, and Ly6G. **b** Unsupervised clustering of ATMs with UMAP where each dot is a single cell colored by cluster assignment. **c** Repartition of ATMs in each cluster per condition. **d** Heatmap of each cell's (column) scaled expression of the top 25 conserved DEGs (row) expressed per cluster, with exemplar genes labeled (right). **e** Violin plots of canonical ATM gene expression by cluster. **f–h** Slingshot analysis of ATM trajectory in mice kept on control diet. UMAP visualization of the pseudotime values with Cluster 5 as starting point (**f**, **g**). Heat map with spline curves fitted to DEGs along a trajectory from ATMs in Cluster 5 to ATMs in Cluster 1 (**h**).

expression of genes such as *ApoE*, *Lyve1*, *Fcna*, and *Folr2* highly expressed by Cluster 1, while antigen-presentation genes were transiently induced in ATMs from cluster 4 (Fig. 1h). An analogous trajectory was found when analyzing ATMs from mice fed HFD overnight (Fig. S1b).

The remaining three clusters represented 7% and 10% of all ATMs from mice kept on CD or fed HFD overnight, respectively. Cluster 6 was characterized by high expression of *Ccr2*, *Plac8*, and *Cx3cr1*. Cluster 7 was distinguished by the expression of *Cd209a*, *Napsa*, *Cd74*, *Flt3*, and *H2-Eb1*, a transcriptional signature associated with classical dendritic cells[28,35]. Cluster 8 was distinguished by the expression of *Trem2* and *Cd9*, similar to

LAMs identified in AT of obese mice[22–24]. This subset of metabolically active ATMs was thus present, albeit in small number, in mice kept on CD and mice fed overnight on HFD (Figs. 1d, e and S1a and Data file S1). Therefore, the ATM compartment is highly heterogenous in lean mice, comprising macrophages with a transcriptional signature indicative of recent differentiation from monocytes and macrophages showing genes associated with tissue residency.

**Establishment of the Lyve1$^+$Tim4$^+$ ATM population is associated with long-term residency.** We used flow cytometry to

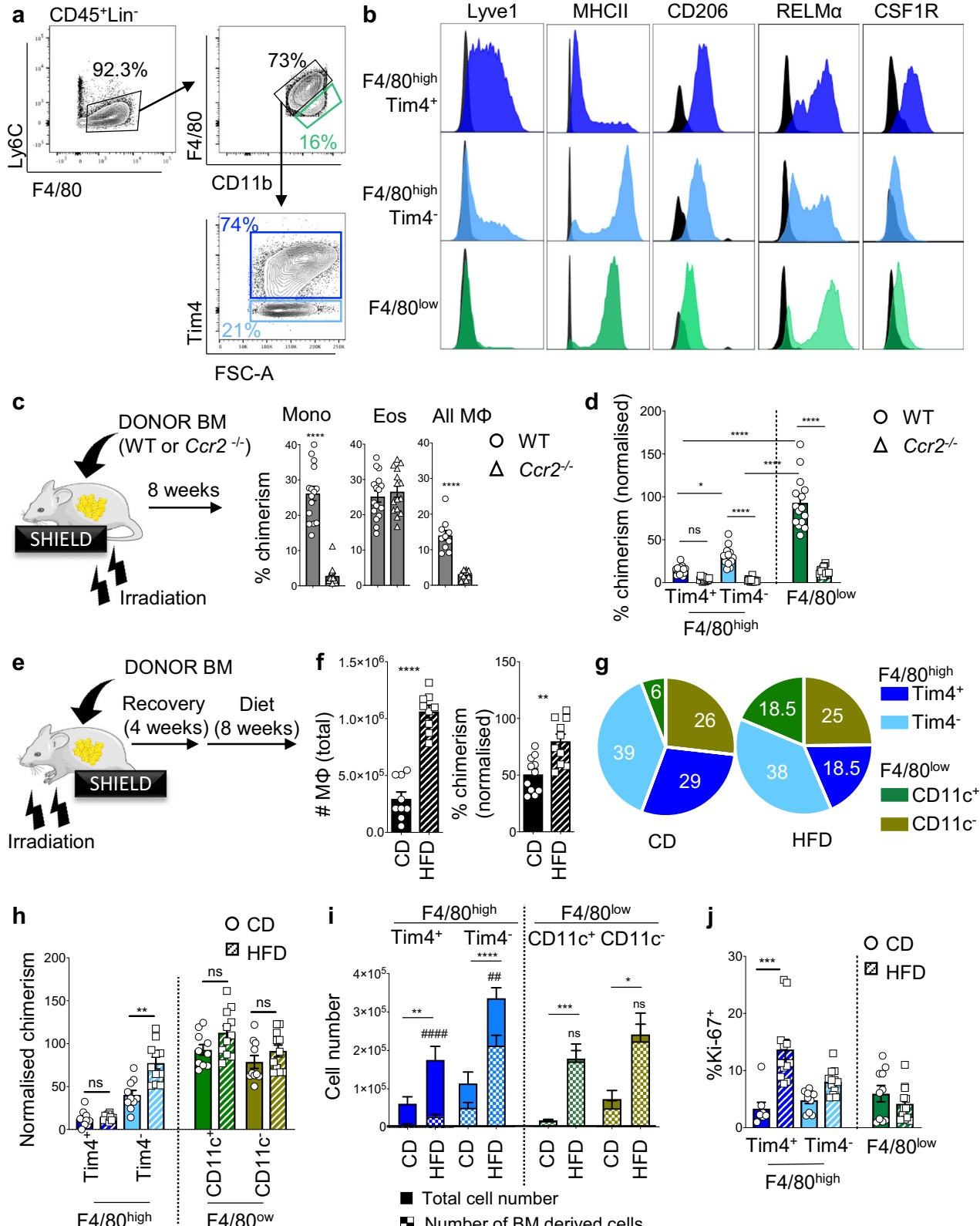

investigate the expression of membrane markers defining Cluster 1. While our scRNA-seq analysis showed relatively low expression of *Timd4* in Cluster 1, previous studies have shown its expression by resident macrophages[25,28] prompting the inclusion of Tim4 in our analysis. Among Lineage− cells, we could identify Ly6C[high] monocytes and define F4/80[high] and F4/80[low] ATM populations. F4/80[high] cells could be further separated into Tim4[+] and Tim4[−]

subsets, whereas F4/80[low] cells were uniformly Tim4[−] (Fig. 2a). F4/80[high]Tim4[+] ATMs had a membrane expression profile compatible with Cluster 1 from our scRNA-seq, with high expression of Lyve1 and CSF1R and low expression of major histocompatibility complex II (MHCII; Figs. 2b, 1e, h, and S1a). In contrast, the majority of F4/80[high]Tim4[−] ATMs lacked Lyve1 and expressed high levels of MHCII suggesting that they corresponded

**Fig. 2 Lyve1+Tim4+ ATMs are resident and persist during obesity. a, b** Gating strategy used to define F4/80$^{high}$Tim4$^+$ (blue), F4/80$^{high}$Tim4$^-$ (cyan), and F4/80$^{low}$ (green) macrophage populations in AT (**a**) and histogram of the fluorescence intensity of Lyve1, MHCII, CD206, RELMα, and CSF1R with fluorescence minus one (FMO) in black (**b**). **c** Hosts were partially irradiated (limbs) and reconstituted with *Ccr2$^{+/+}$* (Wild Type (WT)) or *Ccr2$^{-/-}$* mice. Non-host chimerism (%) among Ly6C$^{high}$ monocytes, eosinophils, and F4/80$^+$ macrophages in the epididymal AT, 8 weeks post reconstitution with WT (circle) or *Ccr2$^{-/-}$* (triangle) BM. **d** Non-host chimerism among ATM subsets, normalized to Ly6C$^{high}$ blood monocyte non-host chimerism. **e–j** Hosts were partially irradiated (head and forelimbs) and reconstituted with WT BM. After recovery, animals were put on CD (solid bar) or HFD (stripped bar) for 8 weeks (**e**). Non-host chimerism was normalized to Ly6C$^{high}$ blood monocyte chimerism. Number and non-host chimerism of the whole F4/80$^+$ ATM populations (**f**) and proportions (%) of ATMs in epididymal AT (**g**). Non-host chimerism among ATM subsets. **i** Total number of ATMs (solid) and BM-derived ATMs (squared pattern). Statistical analyses were performed to compare total ATMs in CD vs HFD (*) and total ATMs vs BM-derived ATMs in HFD (#) (**h**). ATM proliferation measured by percentage of Ki-67$^+$ cells (**j**). Data pooled from $n = 10$ mice per groups from 2 to 3 independent experiments. Error bars show SEM. Kruskal–Wallis test with Dunn's multiple comparisons test or ANOVA with Sidak's multiple comparisons test were applied after assessing normality using D'Agostino and Pearson Normality test. Significant differences are indicated by *$P < 0.05$, **$P < 0.01$, ***$P < 0.001$, ****$P < 0.0001$, $^{##}P < 0.01$, $^{####}P < 0.0001$, ns = non-significant.

to Clusters 2 and 3 (Figs. 2b, 1e, h, and S1a). Both F4/80$^{high}$Tim4$^+$ and F4/80$^{high}$Tim4$^-$ ATM populations expressed high levels of CD206 in agreement with high *Mrc1* expression by Clusters 1–3 (Figs. 2b and S1a). F4/80$^{low}$ ATMs did not express Lyve1, CSF1R, or Tim4 and only low levels of CD206 (Figs. 2b, 1e, h, and S1a). RELMα (encoded by *Retnla*) was expressed by most ATMs. However, differential expression of RELMα between clusters enabled us to discriminate F4/80$^{high}$Tim4$^-$RELMα$^{low/-}$ and F4/80$^{high}$Tim4$^-$RELMα$^{high}$ ATMs corresponding to Clusters 2 and 3, respectively, and to define two subsets in F4/80$^{low}$ macrophages: F4/80$^{low}$RELMα$^{high}$ ATMs potentially corresponding to Clusters 4 and 5 and F4/80$^{low}$RELMα$^{low}$ ATMs corresponding to Clusters 7 and 8 (Figs. 2b and 1e).

Having defined these ATM populations by flow cytometry, we next investigated their replenishment kinetics using AT-protected BM chimeras as described previously in the pleural and peritoneal cavity[36,37]. In brief, after partial irradiation, recipient mice (expressing CD45.1 and CD45.2) were injected with CD45.2 *Ccr2$^{+/+}$* or *Ccr2$^{-/-}$* donor BM. Non-host chimerism of immune cell populations in fat depots was studied 8 weeks later in the blood and tissues (Fig. 2c). As expected, Ly6C$^{high}$ monocytes showed ~30% mixed chimerism in mice who received *Ccr2$^{+/+}$* BM and showed a near complete abrogation of non-host chimerism when mice received *Ccr2$^{-/-}$* BM (Figs. 2c and S2a) in the blood and tissues. In the epididymal AT, the non-host chimerism of eosinophils reached 30%, similar to Ly6C$^{high}$ monocytes and was C-C chemokine receptor type 2 (CCR2) independent (Figs. 2d and S2a). Although ATMs as a whole (CD45$^+$Lin$^-$F4/80$^+$) had a 14% non-host chimerism (Figs. 2c and S2a), further breakdown of the ATM population revealed high heterogeneity in BM dependency. F4/80$^{low}$ ATM subsets were highly BM and CCR2 dependent, with a tissue non-host chimerism of 100% when normalized to Ly6C$^{high}$ blood monocytes, reflecting their constant replenishment by BM monocytes. In contrast, F480$^{high}$Tim4$^+$ ATMs (corresponding to Cluster 1) showed only a low level of non-host chimerism with 14% of Lyve1$^+$Tim4$^+$ ATMs being replaced by BM monocytes after 8 weeks, confirming that the F4/80$^{high}$Tim4$^+$Lyve$^+$ ATM population was maintained in AT over a long period of time with minimal BM monocyte input. The F4/80$^{high}$Tim4$^-$ subset showed intermediate (30%) non-host chimerism at 8 weeks indicating higher contribution of BM monocytes to the maintenance of the F480$^{high}$Tim4$^-$ ATM population compared to the F4/80$^{high}$Tim4$^+$Lyve$^+$ ATM population (Figs. 2d and S2b). The gradual decrease in the incorporation of BM-derived monocytes in ATMs from F4/80$^{low}$ ATMs to F4/80$^{high}$Tim4$^-$ ATMs to F4/80$^{high}$Tim4$^+$ ATMs was in agreement with the trajectory analysis of scRNA-seq data indicating lineage relationship between monocyte-derived ATMs (Clusters 5 and 4 identified as F4/80$^{low}$ ATMs) and F4/80$^{high}$Tim4$^-$ ATMs (Clusters 3 and 2) and

F4/80$^{high}$Lyve1$^+$Tim4$^+$ ATMs (Cluster 1) and with analysis from Silva et al.[25].

Since obesity is characterized by recruitment of ATMs, and a comparative loss of resident ATMs[22], we tested whether obesity led to a change in turnover of Lyve1$^+$Tim4$^+$ resident ATMs. We added CD11c to our flow cytometric analysis, as CD11c has been used extensively to stain inflammatory ATMs in obesity[17–21]. F4/80$^{low}$ ATMs could be further separated into CD11c$^+$ and CD11c$^-$ subsets, whereas F4/80$^{high}$ cells were uniformly CD11c$^-$ (Fig. S2c). We generated protected BM chimeras, which we subjected to 8 weeks of HFD (Fig. 2e). Mice gained significant weight and the total number of ATMs showed a nearly threefold increase in the epididymal AT of mice on HFD compared to mice kept on CD (Figs. S2d and 2f). As expected, there was a significant increase in the proportion and number of F4/80$^{low}$CD11c$^+$ and F4/80$^{low}$CD11c$^-$ ATM subsets (Fig. 2g, i) in the epididymal AT of obese mice. When considered as a whole population, the non-host chimerism of CD45$^+$Lin$^-$F4/80$^+$ ATMs increased by 50% in obese mice (Fig. 2f). The turnover of F4/80$^{low}$CD11c$^+$ and F4/80$^{low}$CD11c$^-$ ATM subsets was of 100% in both control diet and HFD mice, and the turnover of F4/80$^{high}$Tim4$^-$ ATMs rose from 40% in control diet mice to 80% in HFD mice indicating that increased monocyte recruitment contributed to expansion of the ATM pool in obesity (Fig. 2h). In contrast, F4/80$^{high}$Tim4$^+$ resident macrophages showed identical low BM non-host chimerism in mice fed HFD and mice fed a control diet (Fig. 2h). However, their number was twice higher in mice fed HFD compared to mice kept on control diet (Fig. 2i). Analysis of Ki-67 expression showed that the level of proliferation in F4/80$^{high}$Tim4$^+$ ATMs in the epididymal AT of obese mice was significantly higher than that seen in lean mice, consistent with self-autonomous expansion of resident ATM population during obesity (Fig. 2j).

**Lyve-1$^+$Tim4$^+$ ATMs have a unique metabolic profile characterized by high lysosomal activity, high lipid content, and ABCA1 expression.** Having established that the Lyve1$^+$Tim4$^+$ ATM population (Cluster 1) was associated with long-term residence in AT, we analyzed how the AT shaped this population by comparing it to Cluster 4, which is made of ATMs recently derived from monocytes. Pathway analysis of genes with increased expression by Cluster 1 over Cluster 4 showed a very strong enrichment in DEGs involved with lysosomal function, such as *Ctsb, Ctsc, Ctsl, Lgmn, Cd63, Lamp1*, and *Lamp2* (Fig. 3a, b and Data file S2). Flow cytometric analysis confirmed that F4/80$^{high}$Lyve1$^+$Tim4$^+$ ATMs had the highest lysosomal content/activity in steady state compared to the other ATM subsets as assessed by mean fluorescence intensity (MFI) of LAMP2 and lysotracker (Fig. 3c, d). Pathway analysis confirmed enrichment in DEGs involved in endocytosis such as *Cltc, Dab2, Ap2a2,*

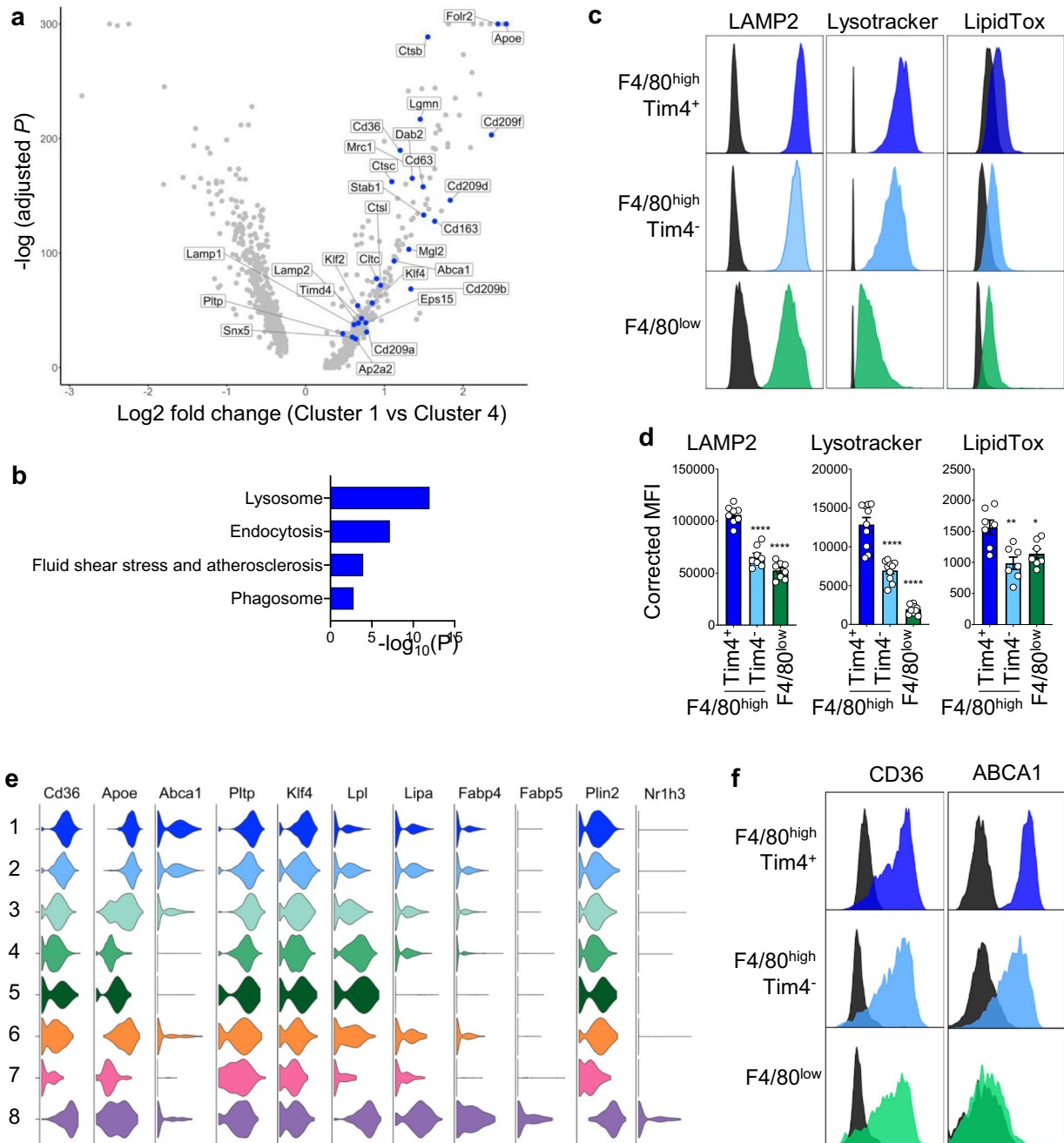

**Fig. 3 Lyve1⁺Tim4⁺ ATMs have high lysosomal and lipid content. a** Volcano plot showing DEGs between Cluster 1 (*Lyve1⁺* resident ATMs) and Cluster 4 (BM-derived ATMs). Examples of DEGs distinguishing Cluster 1 are shown in blue. **b** KEGG pathway analysis on DEGs distinguishing Cluster 1. **c, d** Flow cytometric analysis on epididymal AT showing histogram of the fluorescence intensity of LAMP2, lysotracker, and LipidTox in F4/80^high^Tim4⁺, F4/80^high^Tim4⁻, and F4/80^low^ ATM populations as gated in Fig. 2a with FMO in black (**c**) and quantification of mean fluorescence intensity (MFI) for these staining on the indicated ATM populations (**d**). Data pooled from two independent experiments with n = 8 (LAMP2), n = 10 (lysotracker), or n = 7 (LipidTox) mice per group. Error bars show SEM. ANOVA with Sidak's multiple comparisons test were applied after assessing normality using D'Agostino and Pearson Normality test. Significant differences are indicated by *P < 0.05, **P < 0.01, ****P < 0.0001. **e** Violin plots by cluster of the expression of genes involved in lipid metabolism. **f** Flow cytometric analysis showing histogram of the fluorescence intensity of CD36 and ABCA1 in ATM populations as defined in **c**. Data representative of n = 7 mice per group in two independent experiments.

*Eps15*, and *Snx5* and phagocytosis such as *Cd209a*, *Cd209b*, *Cd209f*, *Cd209d*, *Cd163*, *Stab1*, *Mrc1*, *Timd4*, and *Mgl2* (Fig. 3a, b and Data file S2)[25]. Flow cytometric analysis of neutral lipid content using LipidTox showed that Tim4⁺Lyve1⁺ ATMs had a higher neutral lipid content than F4/80^high^Tim4⁻ and F4/80^low^ ATMs, suggesting that Tim4⁺Lyve1⁺ ATMs were involved in lipid uptake and metabolism at steady state (Fig. 3c, d).

*Lyve1⁺Tim4⁺* resident ATMs expressed high levels of *Cd36*, a receptor enabling the endocytosis of triacylglycerol-rich lipoprotein particles, similar to LAM (Fig. 3e). CD36 membrane expression was confirmed by flow cytometry (Fig. 3f). Contrary to LAM (Cluster 8), *Lyve1⁺* resident ATMs (Cluster 1) did not show a transcriptional signature characteristic of TG metabolism and displayed low expression of *Lpl* and *Lipa*, which catalyze the lipolysis of TG, as well as low expression of *Fabp4* and *Fabp5*,

which mediate FA oxidation (Figs. 3e and S1c, d and Data file S3). However, Lyve1+Tim4+ resident ATMs were distinguished by the expression of *Abca1* (Fig. 3e). *Lyve1+* resident ATMs were also distinguished by the expression of *ApoE*, which mediates reverse cholesterol transport in macrophages[38], and *Pltp,* encoding the plasma phospholipid transfer protein that transfers phospholipids from TG-rich lipoproteins to HDL and the uptake of cholesterol. In contrast to LAMs, *Lyve1+* resident ATMs did not express the transcription factor *Nr1h3* encoding LXRα, which, in macrophages, regulates the transcription of a large repertoire of genes linked to lipid and cholesterol metabolism, such as *Abca1*[39,40]. Dissociation of the expression of *Abca1* and *Nr1h3* in Lyve1+Tim4+ ATMs was further supported by the ImmGen microarray datasets which showed that, in contrast to liver macrophages that highly express both *Abca1* and *Nr1h3*, ATMs expressed high levels of *Abca1* and no *Nr1h3* (Fig. S4). However, *Lyve1+* resident ATMs showed high expression of *Klf4*, a transcription factor inducing *Abca1* expression and cholesterol efflux from endothelial cells[41,42]. Flow cytometric analysis confirmed high membrane expression of ABCA1 by F4/80^highLyve1+Tim4+ resident ATMs, with low expression on F4/80^highTim4− and no expression on F4/80^low ATMs (Fig. 3f). Taken together, these results indicate that Lyve1+Tim4+ resident ATMs have a unique metabolic profile turned toward ABCA1-dependent cholesterol efflux.

**Tim4 and ABCA1 are closely associated with lysosomes in ATMs.** Wholemount immunofluorescence staining confirmed the presence of Tim4+ ATMs showing high lysosomal and neutral lipid content in mouse and human AT. Tim4 and neutral lipid localized to the lysosomes suggesting that Tim4 was actively involved in the uptake and trafficking of lipid from the membrane to the lysosomes (Fig. 4a, b). To further interrogate the human ATM populations, we used flow cytometric analysis, identifying two populations of macrophages within human visceral and subcutaneous AT: CD14+CD16−CD206^highCD64+ macrophages (P3) resembling murine resident ATMs and CD14+CD16+CD206^lowCD64^low ATMs (P2), which appeared to be transitioning from CD16+CD14− monocytes (P1) (Fig. S3a). Both ATM populations expressed Tim4 in visceral and subcutaneous ATs (Fig. S3a–c). Taken together, our results demonstrate that Tim4+ ATMs are resident in AT of mice and humans where they display a metabolically active profile.

In mice, the expression of ABCA1 was concentrated in some areas of the cytoplasmic membrane or intra-cellular membranes that were in contact with lysosomes and Tim4 but did not directly colocalize with these (Fig. 4c). Taken together, these results suggest a close association between Tim4, the processing of lipids in lysosomes, and ABCA1-mediated cholesterol efflux.

**Lyve1+Tim4+ ATMs show rapid transcriptional adaptation following ingestion of HFD.** We next assessed the effect of ingestion of lipids on the transcriptome of the identified clusters. Overnight HFD feeding led to an increase in the proportions of Cluster 5, which represents 1.8% of all ATMs from CD and 7.8% after HFD, and Cluster 4, which represents 18% of all ATMs from CD and 26% after HFD (Fig. 1c). These clusters correspond to the most recent ATMs, which suggest that overnight HFD feeding increased recruitment of monocyte-derived ATMs. However, this was not yet reflected by an increase in the percentage of F4/80^low ATMs that encompass Clusters 4, 5, 7, and 8 (Fig. S2e). The effect of HFD on ATM recruitment can be seen as early as 3 days[43], thus supporting the idea that a high fat meal is sufficient to impact the composition of the ATM compartment.

Analysis of DEGs between mice kept on CD and mice fed overnight with HFD revealed 20 and 40 DEGs in Clusters 1–4,

respectively, and only a limited or null number of DEGs in Cluster 5–8 (Fig. 5a and Data file S3). Pathway analysis on the DEGs induced by the overnight HFD for Clusters 1–4 revealed a unique enrichment in terms associated with lipid response, intra-cellular signaling, and cell metabolism in Lyve1+Tim4+ ATMs (Cluster 1) compared to all other ATM clusters, suggesting that Cluster 1 was readily adapting to increased post-prandial lipid circulation (Fig. 5b and Data files S3 and S4). *Abca1*, *Cd36*, *Hspa1a* (*Hsp70*), and *Malat1* were among the DEGs showing increased expression in HFD vs CD in this cluster. HSPA1a and *Malat1* have both been shown to regulate *Abca1* expression[44,45]. *Nr1h3* was not upregulated in Cluster 1, suggesting that the upregulation of *Abca1* and the lipid response induced in Lyve1+Tim4+ ATMs following exposure to excess lipids differed from the LXR-dependent expression of ABCA1 induced in response to increased cellular cholesterol[39,40] and as shown here by LAMs in Cluster 8 (Fig. 5c).

**Tim4 regulates post-prandial cholesterol transport.** GWAS have highlighted a correlation between *TIMD4* and blood cholesterol in various human cohorts[26,27]. To test the hypothesis that Tim4 regulates circulating cholesterol levels in the blood, we blocked Tim4, using an anti-Tim4 Ig (RMT4-54), in mice fed HFD overnight. As a control, we injected an isotype Ig or chloroquine, which blocks lysosomal function (Fig. 6a). Flow cytometric analysis using fluorescently labeled anti-Tim4 (RMT4-54) showed loss of Tim4 staining on F4/80^high ATMs and an increase in the relative frequency of F4/80^highTim4− ATMs, indicating that Tim4 was successfully blocked on ATMs and these were not depleted by in vivo antibody treatment (Fig. 6b). As expected, ingestion of HFD led to a rise in post-prandial circulating non-esterified free fatty acid (NEFA) and total cholesterol compared to CD (Fig. 6c, d). Increased post-prandial NEFA was independent of Tim4 and chloroquine (Fig. 6c). However, blockade of Tim4 or injection of chloroquine reduced the amount of circulating total cholesterol in mice fed HFD compared to controls that had received Ig, achieving an 87% inhibition in the elevation of post-prandial total cholesterol levels induced by the ingestion of HFD (Fig. 6d). Ingestion of HFD led to an increase in circulating HDLc and non-HDLc, evaluated as total cholesterol minus HDLc. Tim4 blockade specifically abrogated the release of HDLc, similarly to chloroquine and had no effect on non-HDLc (Fig. 6d). Ingestion of lipids induces a transient raise in circulating TG levels, which normalize in a couple of hours. At the time of cull, the levels of circulating TG were not elevated in mice fed a HFD overnight compared to mice kept on CD, indicating that TG levels had already normalized (Fig. 6e). Collectively, these results indicated that Tim4 was involved in the generation of post-prandial HDLc following the ingestion of HFD, implicating Tim4+ macrophages in activation of the reverse cholesterol pathway following ingestion of a lipid-rich meal.

**Tim4 regulates post-prandial increase in ATM lysosomal function.** Flow cytometric analysis of ATMs showed that ingestion of HFD led to increase in lysosomal content of F4/80^high-Tim4+ ATMs as shown by increased MFI of lysotracker in this ATM population in mice fed HFD compared to mice kept on CD (Fig. 6f). To allow detection of Tim4 in mice receiving anti-Tim4 Ig, Tim4 was detected using rat anti-Tim4 Ig plus secondary anti-Rat Ig-647. Increased lysosomal content was only seen in F4/80^highTim4+ ATMs and not in F4/80^highTim4− and F4/80^low ATM populations, implying that lipid ingestion was specifically activating lysosomal function in F4/80^highTim4+ ATMs. Increased lysosomal content following HFD was dependent on Tim4 and blocked by chloroquine, arguing a critical role for Tim4 in increasing lysosomal function following HFD feeding (Fig. 6f).

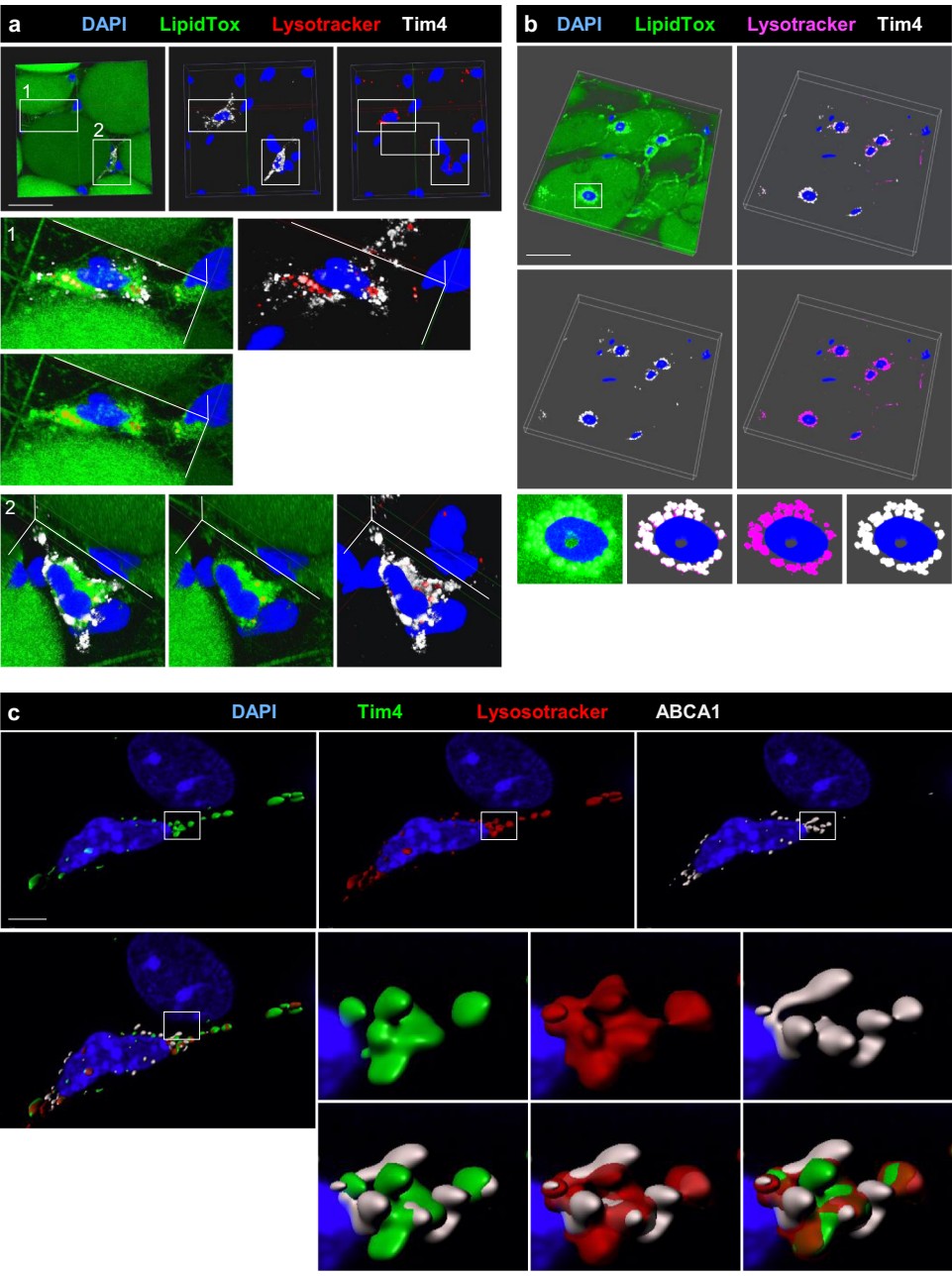

**Fig. 4 Tim4 is closely associated with the lysosomes in murine and human ATMs. a** Confocal imaging and 3D reconstruction of wholemount murine epididymal AT immunofluorescence staining with DAPI (blue), LipidTox green, lysotracker (red), and Tim4 (white). Clipped view showing Tim4+ ATMs found inside the AT are shown in enlargement 1 and 2. Staining representative of n = 8 mice in 2 independent experiments. **b** Confocal imaging and 3D reconstruction of wholemount immunofluorescence staining of human omental AT with DAPI (blue), LipidTox (green), lysotracker (magenta), and Tim4 (white). Enlargement of Tim4+ ATM is shown on the last line of the panel. Staining representative of n = 4 patients. Scale bar 50 μm. **c** Confocal imaging and 3D reconstruction (IMARIS Software) of wholemount murine epididymal AT immunofluorescence staining with DAPI (blue), Tim4 (green), lysotracker (red), and ABCA1 (white). Enlargement are shown on the six images on the lower right-hand side of the panel. Staining representative of n = 4 mice in 2 independent experiments, Scale bar 4 μm.

HFD feeding led to a marked increase in the membrane expression of CD36 on all ATM subsets (Fig. 6g), in agreement with the scRNA-seq data, indicating that ATMs rapidly increase their capacity to uptake lipids upon high fat feeding (Fig. 5c). Intriguingly, the increase in CD36 expression on F4/80^highTim4+ ATMs was potentiated by anti-Tim4 blockade. It is possible that Tim4 blockade limits the internalization of CD36 induced by increased processing of lipids following HFD feeding. The rise in membrane expression of CD36 after HFD was inhibited by chloroquine. This may indicate that increased lipid processing in lysosomes induces a raise in expression of *Cd36* and/or that lysosomes directly regulate the cellular trafficking of CD36 to the membrane. We tested in vitro the role of CD36 in the uptake of lipid particles by F4/80^highTim4+ ATMs using LDL-BODIPY. CD36 blockade abrogated LDL uptake by F4/80^highTim4+ ATMs. In contrast, Tim4 blockade did not prevent LDL uptake but did inhibit increase in lysosomal content induced by LDL. We confirmed by flow cytometry using Annexin V that the lipoprotein particles LDL and chylomicrons were covered with phosphatidylserine, indicating that the interaction

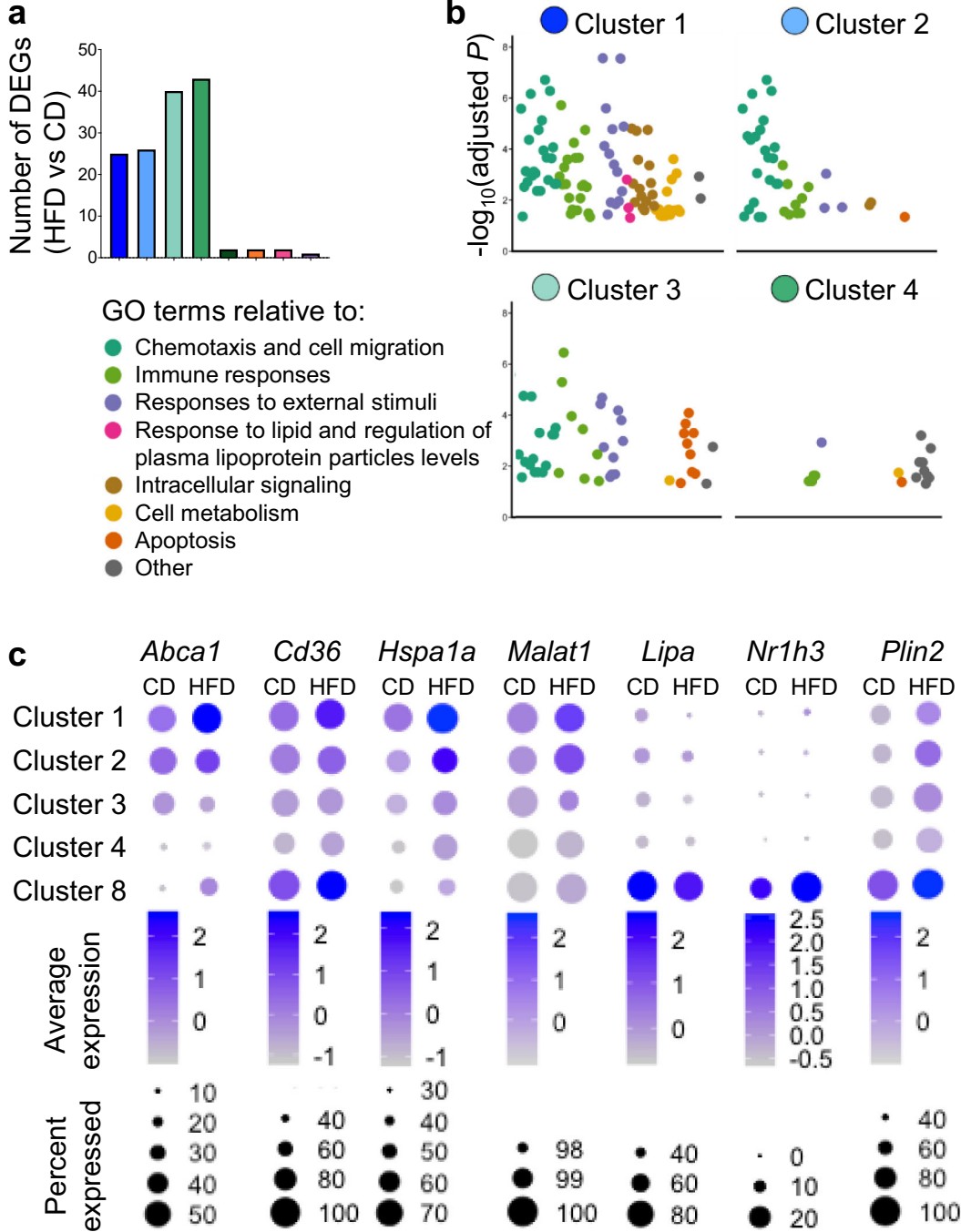

**Fig. 5 Rapid metabolic adaptation of Lyve1⁺Tim4⁺ ATMs to HFD ingestion. a** Bar graph showing the number of DEGs between HFD and CD in the 8 ATM clusters. **b** Pathway analysis on DEGs induced in ATMs from Cluster 1 to Cluster 4 from mice fed overnight with HFD compared to mice kept on CD. Scatter plots show gene ontology (GO) terms categorized per color into families for Cluster 1 to Cluster 4. **c** Density plots showing average gene expression and percentage of cells expressing gene in Clusters 1, 2, 3, 4, and 8 in mice kept on CD and mice fed overnight with HFD.

phosphatidylserine/Tim4 may mediate their trafficking to the lysosomes (Fig. S5c). Taken together, these results indicate that CD36 is critical for LDL uptake and that Tim4 activates lysosomal processing following LDL uptake (Fig. S5a).

Since HFD feeding led to higher *Abca1* expression in F4/80^highTim4⁺ ATMs, we analyzed membrane expression of ABCA1 by flow cytometry. We found that membrane ABCA1 was not increased by HFD feeding and was not altered by Tim4 blockade nor chloroquine (Fig. 6h), indicating that the presence of raised levels of *Abca1* transcripts do not lead to higher membrane expression of ABCA1. However, analysis of *Abca1*

expression in cell-sorted F4/80^highTim4⁺ ATMs confirmed that HFD led to increased *Abca1* expression and revealed that this was dependent on Tim4 as *Abca1* expression was decreased by Tim4 blockade (Fig. 6i). Analysis of the expression of *Nr1h3* in cell-sorted F4/80^highTim4⁺ ATMs showed that HFD feeding did not lead to increased expression of *Nr1h3*, whose expression remained very low ($\Delta Ct > 6$ for *Nr1h3* compared to $\Delta Ct \sim 1$ for *Abca1*) confirming that NR1H3 does not regulate *Abca1* expression in this ATM population (Fig. 6i).

Finally, we confirmed by flow cytometric analysis that overnight HFD feeding led to increased intra-cellular HSP70

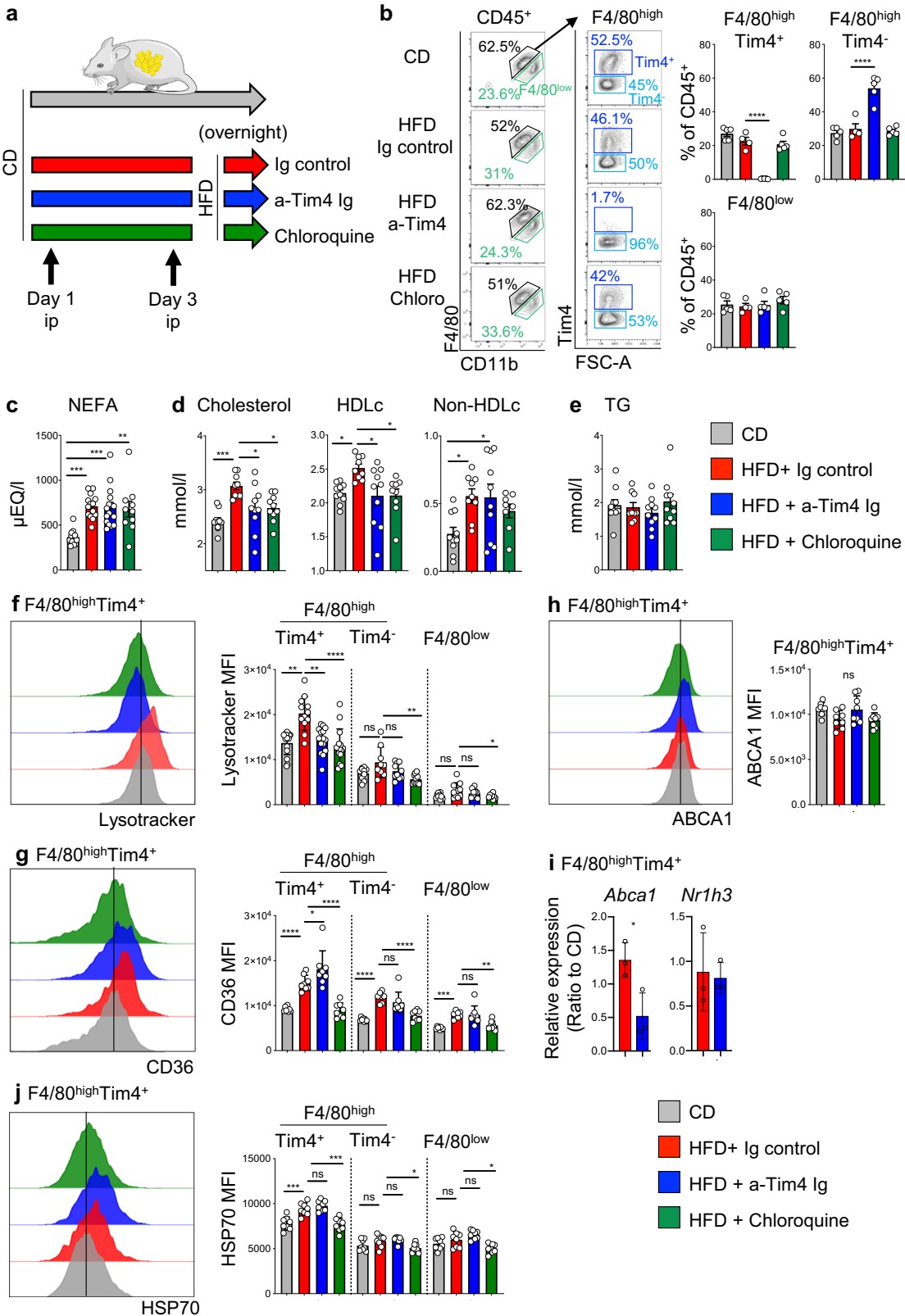

expression by F4/80^highTim4^+ ATMs. This increase was not affected by Tim4 blockade but was inhibited by chloroquine (Fig. 6j). Taken together, these results indicate that overnight HFD feeding leads to a rapid increase in CD36 expression and

lysosomal content in F4/80^highTim4^+ ATMs, expanding their capacity to uptake and process lipids. While Tim4 is not required to potentiate CD36 and HSP70 expression, it is critical to increase F4/80^highTim4^+ ATM lysosomal function after HFD feeding.

**Fig. 6 Blocking Tim4 impairs post-prandial cholesterol metabolism. a** Mice were kept on CD or fed with HFD overnight. Mice fed HFD were injected i.p. with Ig control or blocking anti-Tim4 Ig or chloroquine, 72 h and on the day prior to the overnight HFD. **b** Flow cytometric analysis on ATMs showing F4/80[high]Tim4[+], F4/80[high]Tim4[−], and F4/80[low] ATM populations and quantification of their relative proportions in mice kept on CD (gray bar) and mice fed overnight with HFD and injected with Ig control (red), anti-Tim4 Ig (blue), or chloroquine (green). Data are representative of 2 independent experiments with *n* = 5 mice per groups. **c–e** Circulating levels of NEFA (**c**), total cholesterol, HDLc, and non-HDLc (**d**), and TG (**e**) in mice kept on CD (gray bar) and mice fed overnight with HFD and injected with Ig control (red), anti-Tim4 Ig (blue), or chloroquine (green). **f–h, j** Flow cytometric analysis showing histogram of lysotracker (**f**), CD36 (**g**), ABCA1 (**h**), and HSP70 (**j**) fluorescence intensity in F4/80[high]Tim4[+] ATMs and quantification of MFI in the indicated ATM populations from mice kept on CD (gray bar) and mice fed overnight with HFD and injected with Ig control (red), anti-Tim4 Ig (blue), or chloroquine (green). To allow detection of Tim4 in mice receiving anti-Tim4 Ig, Tim4 was detected using rat anti-Tim4 Ig plus secondary anti-Rat Ig-647. Data are pooled from *n* = 8–15 mice per group from 2 to 3 independent experiments. **i** Ratio of relative amounts of *Abca1* and *Nr1h3* expressed by F4/80[high]Tim4[+] ATMs isolated from mice kept on CD or fed with HFD overnight and treated with Ig control or blocking anti-Tim4 Ig as in (**a**). Data are pooled from *n* = 3 biological replicates per group. Error bars show SEM. Kruskal–Wallis test with Dunn's multiple comparisons test or ANOVA with Sidak's multiple comparisons test were applied after assessing normality using D'Agostino and Pearson Normality test (**b–h, j**). Two-tailed Student's *T* test was applied in (**i**). Significant differences are indicated by **P* < 0.05, ***P* < 0.01, ****P* < 0.001, *****P* < 0.0001, ns = non-significant.

**Tim4[+] liver Kupffer cells and peritoneal cavity macrophages are not required for increased circulating post-prandial HDLc.** Tim4 is expressed by resident macrophages of almost all tissues[28,29] and therefore the effects of Tim4 blockade on HDLc could reflect effects on other Tim4[+] macrophages. In particular, liver Kupffer cells and peritoneal macrophages represent two important reservoirs of Tim4[+] resident macrophages and so we first compared the lysosomal content of Tim4[+] ATMs with Tim4[+] Kupffer cells and Tim4[+] resident peritoneal cavity macrophages. In mice kept on control diet, we found that Tim4[+] ATMs had a higher lysosomal content than Tim4[+] Kupffer cells and Tim4[+] peritoneal macrophages, suggesting that Tim4[+] ATMs had a higher metabolic activity (Fig. 7a). ABCA1 membrane expression mirrored the lysosomal content of these cell types, with highest expression in ATMs and lowest expression in Kupffer cells and peritoneal macrophages (Fig. 7b). In resident peritoneal macrophages, which can be subdivided as Tim4[+] and Tim4[− 37], expression of ABCA1 was markedly lower in Tim4[−] macrophages, suggesting that expression of ABCA1 is linked to Tim4 (Fig. S5D). Next, we used clodronate liposome-mediated cell depletion to rule out a role for Tim4[+] macrophages in liver and peritoneal cavity in mediating the effects of anti-Tim4 blockade. We found that Tim4[+] Kupffer cells and peritoneal macrophages were very efficiently depleted by intraperitoneal (i.p.) delivery of clodronate liposomes. However, Tim4[+] ATMs were only partially depleted (~40%; Fig. 7c, d). To test whether Tim4[+] Kupffer cells and Tim4[+] large peritoneal macrophages were required to raise the levels of post-prandial cholesterol following ingestion of HFD, mice received one i.p. injection of clodronate liposomes 24 h prior to being given HFD overnight (Fig. 7e). As expected, ingestion of HFD led to increased levels of circulating NEFA, which was not affected by injection of clodronate liposomes (Fig. 7f). In addition, injection of clodronate liposomes did not impair the rise in the levels of post-prandial circulating total cholesterol and HDLc following ingestion of HFD, in support of a minor role for peritoneal macrophages and Kupffer cells in the regulation of post-prandial cholesterol levels (Fig. 7g). As in the previous experiment, the levels of TG were unchanged (Fig. 7h). Taken together, these results suggest that other resident Tim4[+] macrophages such as Tim4[+] ATMs, that are only partially depleted by clodronate liposomes, are sufficient to raise post-prandial HDLc levels after ingestion of HFD.

## Discussion

AT is an ever-changing niche, adapting to food intake and fluctuation in energy needs. In this study, we focused on defining the role of resident ATMs in lean mice challenged with a high-fat meal. Using scRNA-seq, we demonstrated that AT residency is associated in ATMs with the expression of Lyve1, Tim4, and ABCA1 and the acquisition of high endocytic and lysosomal capacity. Challenge with a high-fat meal led to specific transcriptional activation of resident ATMs characterized with increased *Abca1* expression and lysosomal function. Blocking Tim4 led to inhibition of lysosomal function in ATMs as well as dysregulation of post-prandial cholesterol transport, with decreased levels of circulating HDLc. We ruled out a role for liver Kupffer cells and peritoneal macrophages, two important reservoirs of Tim4[+] resident macrophages, using clodronate liposomes. We thus propose a model whereby, after ingestion of lipids, Tim4[+] Lyve1[+] resident ATMs allow the reverse transport of cholesterol to HDL, recycling post-prandial cholesterol from chylomicron remnants. Tim4 links CD36-mediated uptake of phosphatidylserine-covered lipoprotein to lysosomal processing and induces increased *Abca1* expression. By facilitating in situ and in real time reverse cholesterol transport, resident ATMs limit the circulation of chylomicron remnants, which are potentially harmful (Fig. 8).

While we could rule out a role for liver macrophages in the regulation of post-prandial cholesterol transport using clodronate liposomes, our results did not allow us to quantify the relative importance of Tim4[+]Lyve1[+] resident macrophages of the AT compared to other tissues in the regulation of post-prandial HDLc. The existence of Lyve1[high] MHCII[low] resident macrophages has been reported in all tissues, and their frequency seems to be dependent on tissue type[22,25,28,32,46]. Further studies will investigate ABCA1 expression and the metabolic profile of these resident macrophages in various tissues. Expression of ABCA1 on resident macrophages is probably differentially regulated depending on tissue type and the ability of this tissue to induce the release of FAs from chylomicrons. AT represents a major site for processing and storage of dietary lipids suggesting that tissues which induce the largest release of TG from chylomicrons are associated with a resident population of macrophages specializing in the initiation of reverse cholesterol transport via HDL.

Studies showed that reverse cholesterol transport in macrophages is dependent on LXRα, which is induced by excess cholesterol in cells[39,40]. Here we found that ATMs do not express *Nr1h3* which suggests that the mechanism controlling *Abca1* expression in ATMs is different and may be regulated by Tim4 as our Tim4 blockade experiment in mice fed overnight HFD suggests. Tim4[+] ATMs express *Klf4*, HSP70, and Malat-1, which have been linked to *Abca1* expression and reverse cholesterol transport. Another intriguing finding is that resident ATMs do not express *Lipa*, which allows the digestion of cholesterol ester accumulated in lipid droplets in macrophages. However, regulation of post-prandial HDLc levels was inhibited by chloroquine, an inhibitor of lysosome function, and confocal microscopy showed a close association between Tim4, neutral lipids, and lysosomes, indicating that the process was dependent on

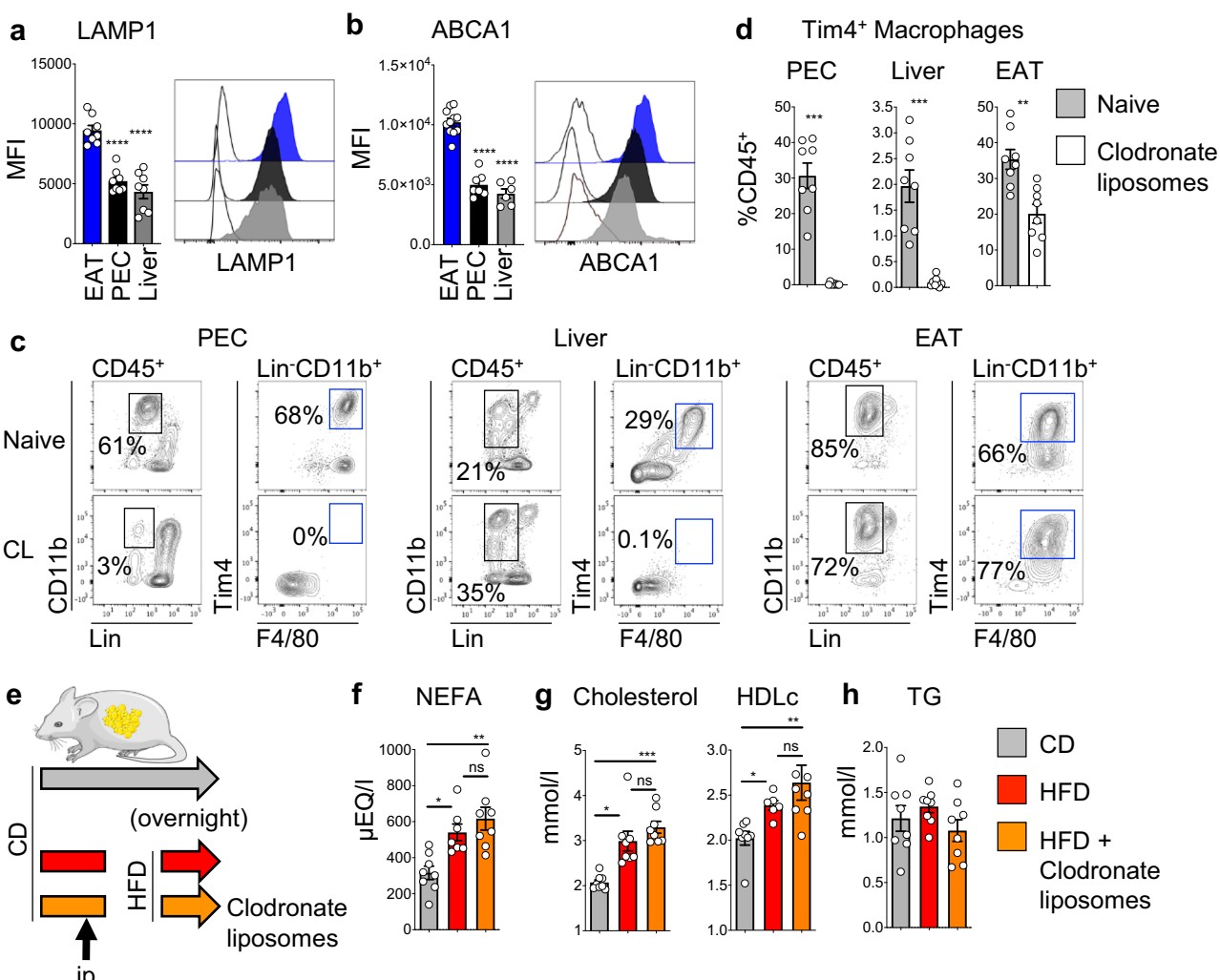

**Fig. 7 Depletion of liver and peritoneal macrophages does not alter post-prandial cholesterol metabolism. a**, **b** Quantification of lysotracker (**a**) and ABCA1 (**b**) mean fluorescence intensity (MFI) and representative histogram of the fluorescence intensity for Tim4+ macrophages in the epididymal AT (EAT), peritoneal cavity exudate cells (PEC), and liver (gating shown in **c**). **c**, **d** Flow cytometric analysis showing gating strategy (**c**) used to quantify the percentage of Tim4+ macrophages (**d**) present in the PEC, liver, and EAT of naive mice (gray bar) and mice that received clodronate liposomes 24 h prior to analysis (white bar). Data are pooled from n = 8 mice per group from 2 independent experiments. **e** Mice were kept on chow diet or fed with a HFD overnight. One group of mice fed a HFD were injected i.p. with clodronate liposomes 24 h prior to the overnight HFD. **f–h** Circulating levels of NEFA (**f**), total cholesterol and HDLc (**g**), and TG (**h**) in mice kept on CD (gray bar), mice fed overnight with HFD (red bar), and mice fed overnight with HFD and injected with clodronate liposomes (orange). Data are pooled from n = 7 or 8 mice per group from 2 independent experiments. Error bars show SEM. Kruskal–Wallis test with Dunn's multiple comparisons test or ANOVA with Sidak's multiple comparisons test were applied after assessing normality using D'Agostino and Pearson Normality test. Significant differences are indicated by *$P < 0.05$, **$P < 0.01$, ***$P < 0.001$, ****$P < 0.0001$, ns = non-significant.

lysosomes. In tumor-associated macrophages, Tim4 was shown to be dispensable for the uptake of apoptotic tumor cells but to be critical for lysosomal activation and the degradation of ingested tumor cells[47]. Our in vitro experiments indicate that, in ATMs, a similar mechanism is involved in the transport of cholesterol-rich lipoprotein particles, whose uptake is dependent on CD36 and their transport to the lysosomes mediated by Tim4. The fact that ABCA1 was concentrated in some membrane areas in contact with lysosomes and Tim4 could suggest that, in Tim4+ ATMs, a mechanism enables excess cholesterol from cholesterol-rich lipoprotein particles to be transferred from lysosomes to the cytoplasmic membrane and ABCA1 for export. Further studies are required to dissect the molecular mechanisms involved in the regulation of *Abca1* expression and cholesterol efflux in resident ATMs.

GWAS identified genetic variants of *Timd4* associated with dyslipidemia. Here we provide a potential physiological

mechanism to explain this association. Experimental blockade of Tim4 in *Ldlr−/−* mice was shown to worsen atherosclerosis. Tim4 blockade led to decreased efferocytosis and increased T cell activation but had no influence on circulating cholesterol levels[48]. However, *Ldlr−/−* mice are highly dyslipidemic with 1000 mg/dl (~55.5 mmol/l) of circulating cholesterol compared to the WT mice used in our study with a circulating cholesterol level of 2.5 mmol/l. It is likely that the pronounced dyslipidemia associated with *Ldlr−/−* mice masked the effect of Tim4 on regulation of cholesterol levels. Contrary to mice, which have a circulating lipoprotein profile dominant in HDLc, humans have a LDLc dominant profile and are prone to atherosclerosis[49]. Future work is thus required to determine whether the function of Tim4 and ATM macrophages on the regulation of post-prandial cholesterol can be leveraged to raise HDLc levels in humans. Since the Framingham Heart Study in the 1960s, which was the first to report the strong inverse association between cardiovascular risk and

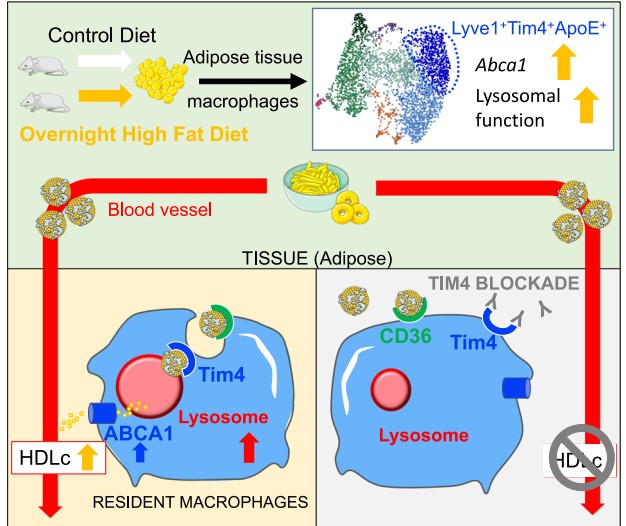

**Fig. 8 Tim4 links regulation of post-prandial cholesterol transport and ABCA1$^{+}$ ATM metabolism.** A high-fat diet (HFD) meal leads to increase in *Abca1* expression and lysosomal function in Tim4$^{+}$ resident adipose tissue macrophages (ATMs). CD36 mediates lipoprotein particle uptake by ATMs, and Tim4 is required for lysosomal activation following uptake. Blocking Tim4 prevents lysosomal activation in ATMs and inhibits post-prandial HDLc release.

plasma HDLc, the therapeutic potential of raising HDLc levels has been lessened by the failure of clinical studies to show that raising HDLc levels improves cardiovascular disease outcome. However, it has become apparent that HDLc levels do not necessarily reflect efficacy of reverse cholesterol transport[7–9]. Our study highlights the importance of understanding the dynamics of cholesterol transport following meals and the role of tissue macrophages in this process. Further studies may uncover new pathways that could be targeted to modulate the efficacy of reverse cholesterol transport for the treatment and prevention of cardiovascular diseases.

## Methods

**Design.** We performed phenotypic, transcriptomic, and functional analysis of ATMs macrophages from mice kept on CD and mice given HFD overnight to characterize resident ATMs and the changes induced by a high fat meal. We used tissue-protected BM chimeric mice to assess the replenishment kinetics of ATMs in lean and obese mice. We assess the role of Tim4 and lysosomal function in the regulation of post-prandial cholesterol levels using blocking anti-Tim4 Ig and chloroquine in mice given HFD overnight. To assess the requirement of liver and peritoneal macrophages in the regulation of post-prandial cholesterol level, we used i.p. clodronate liposomes. The number of experiments performed is indicated in each figure legend.

**Animals.** All experiments were done in compliance with all relevant ethical regulations under a project license granted by the UK Home Office and were approved by the University of Edinburgh Animal Welfare and Ethical Review Board. All individual experimental protocols were approved by a named veterinarian surgeon prior to the start of the experiment. Experiments were performed using male C57BL/6 (C57BL/6JOlaHsd) aged 8–12 weeks. All animals were bred and housed at 22–23 °C on a 12-h light/dark cycle with free access to water and food under specific pathogen-free conditions at the University of Edinburgh Animal Facilities.

Mice were kept on control diet (11 kcal%Fat and corn starch, Research diet, D12328i). For overnight high fat feeding, mice were given HFD (58Kcal%Fat and sucrose, Research diet, D12331i) at 4 p.m. and were culled the next morning at 9 a.m. For Tim4 blocking, mice were i.p. injected, 72 h and on the day prior to the overnight HFD, with 100 μl of phosphate-buffered saline (PBS) containing either 200 μg of anti-Tim4 IgG2a (clone RMT4-53, Rat IgG2a, BE0171, BioXCell) or 200 μg of rat IgG2a control (BE0089, BioXCell). Mice were injected i.p. with 1 mg of chloroquine (Sigma) in 200 μl of PBS 72 h and on the day prior to the overnight HFD. Mice were injected with 200 μg clodronate liposomes (Liposoma, the Netherlands, https://clodronateliposomes.com) in 200 μl PBS 48 h prior to overnight HFD.

Chimeric mice were generated as described by Bain et al.[37]. Sedated 8-week-old C57BL/6J CD45.1$^{+}$CD45.2$^{+}$ mice were exposed to a single dose of 12 Gy γ-irradiation. Either the upper half of the body or the lower half of the body was exposed to irradiation while a 2-inch lead shield was protecting the lower abdomen in order to preserve AT depots from irradiation. Then mice were given intravenously 2 × 10$^6$–5 × 10$^6$ BM cells, obtained from CD45.2$^{+}$ C57BL/6J (WT) or *Ccr2$^{-/-}$* animals. After a minimum of 8 weeks recovery, blood and tissues were collected for flow cytometric analysis. For long-term high fat feeding, mice were kept on HFD (58Kcal%Fat and sucrose, Research diet, D12331i) for 8 weeks.

**Murine tissue preparation.** Murine gonadal AT were enzymatically digested with 1 mg/ml Collagenase D (Roche) for 35 min at 37 °C in RPMI 1640 (Sigma) containing 1% fetal bovine serum (Sigma). Peritoneal exudate cells were isolated by flushing murine peritoneal cavities with RPMI 1640 (Sigma). The liver was perfused before dissection with 5 ml of RPMI 1640 (Sigma) injected through the portal vein. The tissue was cut into small pieces and homogenized using the gentleMACS dissociator (Miltenyi) in buffer containing Collagenase 2 (Sigma 0.425 mg/ml), Collagenase D (Roche 0.625 mg/ml), Dispase (Gibco 1 mg/ml), and DNase (Roche 30 μg/ml). After 20-min incubation at 37 °C, the tissue was homogenized further using the dissociator. Red blood cells were lysed using red blood cell lysis buffer (Sigma).

**Human subjects.** The human study was done in compliance with all relevant ethical regulations, following approval by the East of Scotland Research Ethics Service REC 1 (15/ES/0094), with all patients providing written informed consent prior to any study procedures. Paired human subcutaneous and visceral AT samples (*n* = 4) were obtained from subjects undergoing elective abdominal surgery at the Royal Infirmary of Edinburgh. Samples were put in PBS on ice and used immediately for the respective experiments.

**Human sample preparation.** Human AT was weighed and ±0.500 g of tissue was digested using 2 mg/ml Collagenase I (Worthington) in PBS (Invitrogen/sigma) 2% bovine serum albumin (BSA, Sigma), samples were disrupted using an Octolyser (Miltenyi), incubated at 37 °C with intermittent shaking for 45 min, subjected to a second Octolyser dissociation step, ions were chelated by addition of EDTA (0.5 M, Sigma), samples were filtered through a 100-μm filter (BD), and washed with 20 ml of 2% BSA PBS prior to centrifugation at 1700 rpm for 10 min. The cell pellet was resuspended in 2 ml of PBS 2% BSA for flow cytometric analysis.

**Flow cytometry.** Murine cells were stained with LIVE/DEAD (Invitrogen), blocked with mouse serum and anti-murine CD16/32 (clone 2.4G2, Biolegend), and stained for cell surface markers (see Table S1 for a list of antibodies used). Where lysotracker was used, cells were incubated in RPMI with lysotracker (ThermoFisher, 1/2000) for 30 min at 37 °C and washed in fluorescence-activated cell sorting (FACS) buffer prior to staining for surface markers. For LipidTox staining, cells were first fixed in Neutral buffered formalin (NBF; 10%, Sigma), then stained with LipidTox (ThermoFisher, 1/200) for 30 min at room temperature. Human samples were blocked with serum, stained for cell surface markers (see Table S1 for a list of antibodies used), and DAPI was added to the cells prior to acquisition. All samples were acquired using a FACSDiva software 6.3.1, BD Biosciences software, and analyzed with the FlowJo 10 software (Tree Star). For BODIPY LDL uptake experiments, AT cells were preincubated with anti-CD36 IgA (clone CRF D-2712, BD Pharmingen, 22.5 μg/ml) or anti-Tim4 IgG2a (clone RMT4-53, BioXCell, 22.5 μg/ml) or IgG2a control (clone BE0089, BioXCell, 22.5 μg/ml) prior to incubation with BODIPY FM LDL (10 μg/ml, Invitrogen) for 1 h.

**Wholemount immunofluorescence staining and microscopy.** Human and mouse omentum samples were first incubated with lysotracker (ThermoFisher, 1/1000) for 30 min at 37 °C. Tissues were then fixed for 1 h on ice in 10% NBF (Sigma) prior to staining at room temperature with primary antibodies and LipidTox (1/100, ThermoFisher) for 2 h in PBS 0.5% BSA 0.5% Triton. Antibodies used are listed in Table S1. After mounting with Fluoromount G, confocal images were acquired using a Leica SP8 laser scanning confocal microscope using the Leica LAS X software. Three-dimensional (3D) reconstruction was created using the LAS-X-3D (Leica) v3.5.7.23225 and IMARIS software (2018).

**Cell sorting and quantitative real-time PCR (qRT-PCR).** Cells were stained for cell surface marker and sorted using a FACS Aria Fusion directly in 350 μl RLP buffer before RNA extraction using the RNeasy Plus Micro Kit (Qiagen) according to the manufacturer's instruction. Complementary DNA for mRNA analysis was synthesized from total RNA using the High-Capacity RNA-to-cDNA Kit (ThermoFisher). *Abca1* and *Nr1h3* expression was assessed using TaqManGene Expression Assay (Mm00442646_m1 and Mm00443451_m1) by qRT-PCR (Life Technologies) and normalized to glyceraldehyde-3-phosphate dehydrogenase (*Gapdh*, Mm9999995_g1). Means of triplicate reactions were represented for *n* = 3 biological samples per condition.

**Data pre-processing of droplet-based scRNA-seq data.** CD45$^+$TCRb$^-$CD19$^-$ SiglecF$^-$Ly6G$^-$CD11b$^+$F4/80$^+$ ATMs pooled from the epididymal fat pad of three mice kept on CD or pooled from three mice fed HFD overnight were FACS-sorted using a FACS Aria Fusion and processed using the 10× Chromium (10× Genomics) platform following the recommended protocol for the Chromium Single Cell 3' Reagent Kit. Libraries were run on the NovaSeq S1 for Illumina sequencing. Sequence reads were processed and aligned to the mm10 reference genome (Ensembl 93) using the Cell Ranger v3.0.2 Single-Cell Software Suite from 10× Genomics. Initial quality control (QC) was performed separately for the "CD" and "HFD" conditions, excluding genes if there were expressed in fewer than three cells, and excluding cells based on the following criteria: those expressing <200 or >1.5 times the inter-quartile range more than the upper quantile of genes ($n$, CD = 52, HFD = 47), those with a mitochondrial gene proportion of over 10% of total UMI counts ($n$, CD = 2043, HFD = 1120), or those with a UMI count-to-gene ratio >7 ($n$, CD = 0, HFD = 2). Gene expression was normalized by cell based on its total expression, before being multiplied by a scale factor of 10,000 and log-transformed. A list of 2000 variable genes were generated using the "vst" method of the *FindVariableFeatures* function in the Seurat R package version 3.1.1[50]. Following these QC steps, transcription profiles of 2364 ATMs for the CD condition and 1994 ATMs for the HFD condition, with a median number of genes per cell of 1633 and 1173, respectively, were obtained.

**Sample integration.** Samples from the CD and HFD conditions were combined using anchor-based integration as described in ref. [50], choosing 10 CCA dimensions for *FindIntegrationAnchors*. Following integration, cells were assigned a cell cycle score using the *CellCycleScoring* function from Seurat. UMI variation, percentage of mitochondrial count variation, and cell cycle were regressed against the corrected normalized data using a linear regression. Residuals from this model were centered and scaled by subtracting the average expression of each gene, followed by dividing by the standard deviation of each gene.

**Dimensionality reduction, clustering, differential expression analysis, and data visualization.** The Seurat R package was used to perform all dimensionality reduction, clustering, and differential gene analysis. Shared nearest neighbor clustering was performed on the integrated data using between 1 and 10 principal components (PCs), as determined by the dataset variability shown in the principal component analysis (PCA). The resolution was optimized based on the resulting number of clusters. All differential expression analysis was conducted using a linear regression model on uncorrected normalized data. Conserved differential genes were calculated using the *FindConservedMarkers* function for each cluster individually. Cells from each cluster were compared to all other cells for each test. *FindMarkers* was used to identify DEGs between the Healthy and High Fat cells per cluster. For identification of differential genes between two specific clusters, *FindMarkers* was also utilized using sample (Healthy vs High Fat) as a latent variable. Only genes with at least 0.25 log-fold change and expressed in at least 25% of cells in the evaluating cluster were considered for all tests.

All violin plots, volcano plots, feature plots, UMAPs, and heatmaps were generated using the *Seurat*, *ggplot2*, and *pheatmap* R packages. The sample number of PCs were used for construction of the UMAP as were used previously for clustering. Violin and features plots visualize the uncorrected normalized data. Average fold change and adjusted $P$ value from the differential gene expression comparison shown were used for volcano plots. Adjusted $P$ values $> 1e^{-300}$ were set to $1e^{-300}$ for the purposes of plotting. For heatmap generation, the uncorrected data were scaled in the same manner as above, and the resulting uncorrected scaled expression data was used for heatmap visualization.

**Removal of contaminating clusters.** Clusters annotated as endothelia (*Pecam1*, *Kdr*, *Flt1*), peritoneal macrophages (*Gata6*), or those with a median number of genes <1000 were excluded from further analysis. The resulting data were re-integrated, re-scaled, and re-clustered following the same procedure as described above.

**Trajectory inference.** Cells coming from mice kept on CD and mice fed overnight HFD were separated and the uncorrected data re-scaled followed by PCA analysis. Lineage inference was performed using a cluster-based minimum spanning tree from the *slingshot* R package[51] on PCs 1:10. The same cluster annotations used for the integrated data were used in the trajectory inference, with Cluster 5 defined as the starting cluster. Pseudotime values were visualized on previously generated UMAPs containing only cells from CD or HFD condition. A new set of 2000 variable features were identified and regressed on the pseudotime values using a general additive model. Cubic smoothing splines were fitted to the scaled expression of selected top DEGs along the pseudotime trajectory using the *smooth.spline* (df = 3) function from the *stats* R package and were plotted as a heatmap with range clipped from −2 to 2.

**Gene ontology.** Pathway analysis was performed for each cluster on the DEGs overexpressed by ATMs from mice on overnight HFD over ATMs from mice kept on CD using g:Profiler. Gene ontology biological process results were categorized into families and ordered sequentially per family using the numerical value of the term identification. Terms were then labeled from 1:$n$, and this sequential numbering used to separate terms on a scatter plot to show −log-adjusted $P$ value.

**Statistical analysis.** No randomization and no blinding was used for the animal experiments. All data were analyzed using Prism 7 (GraphPad Prism, La Jolla, CA). Statistical tests performed for each dataset are described within the relevant figure legend.

**Reporting summary.** Further information on research design is available in the Nature Research Reporting Summary linked to this article.

## Data availability

All relevant data are available from the authors upon reasonable request. ScRNA-seq datasets have been deposited at GEO: GSE168278, and the processed scRNA-seq analysis files are provided in Supplementary Data files 1–5. Source data are provided with this paper.

## Code availability

R scripts for performing the main steps of analysis are available from the corresponding authors on reasonable request.

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

## Acknowledgements

We are extremely grateful to Central Bioresearch Services, especially Gidona Goodman, Chris Flockhart, and Will Mungall for their technical support. Flow cytometric and confocal microscopic data were generated with support from the QMRI flow cytometry and cell-sorting facility, the CALM facility, and CRM-Imaging, University of Edinburgh. We thank Matthieu Vermeren for his support with imaging. We thank Karen Chapman and Sonja Vermeren for their kind support and guidance. We thank Lynne Ramage for her technical assistance and we acknowledge the financial support of NHS Research Scotland (NRS) through the Edinburgh Clinical Research Facility. Figures 1a, 2c, e, 6a, 7e, and 8 were adapted and compiled using images from Servier Medical Art. This work was supported by a Medical Research Council (MRC) UK Grant to C.B. (MR/M011542/1) and S.J.J. (MR/L008076/1). L.D. was supported by a Senior Kidney Research UK Fellowship (SF_001_20181122). P.R. was supported by an MRC Clinician Scientist Fellowship (MR/N008340/1). R.H.S. was supported by a fellowship from the Chief Scientist Office (SCAF/17/02). N.C.H. was supported by a Wellcome Trust Senior Research Fellowship in Clinical Science (219542/Z/19/Z).

## Author contributions

M.S.M.: investigation, formal analysis, writing—original draft, and visualization. P.S., J.R.P., L.H.J-J., C.C.B., P.R., Z.M., L.D., M.R.D., R.H.S., N.C.H., and S.J.J.: investigation, analysis, and review. C.B.: funding acquisition, supervision, conceptualization, investigation, project administration, formal analysis, writing—original draft, and visualization.

## Competing interests

The authors declare no competing interests.
