## [Peer Review File · Nature Communications]

Reviewers' Comments:

Reviewer #1:

Remarks to the Author:

Magalhaes et al describe here the heterogeneity of adipose tissue macrophages in normal mice vs mice fed overnight with HFD to study the role of macrophages in regulating post-prandial cholesterol transport. They notably identify an adipose tissue resident macrophage population expressing Tim4 and ABCA1. The study is concise and of interest but however some important experiments or analysis are missing to provide a complete and convincing study.

The data presented in Figure 4 represent the most important part of the study (comparing changes in expression profile in normal mice vs mice fed overnight with HFD) but are not clearly presented. It is not clear to the reviewer if CD versus overnight HFD lead to significant and reproducible changes in cell composition as well as in gene expression profile across the defined clusters. It would be useful to display the data for HFD and compare them point by point / cell population by cell population to CD. As the point here is to show the adaptation of cluster 1 after HFD ingestion so the comparison between cluster 1 and 3 is useless (already done in the previous figures). Authors should insist on CD to HFD comparison here! and show differentially expressed genes etc...

The authors show more ABCA1 mRNA in cluster 1 in CD vs HFD. But surprisingly they do not compare its expression by flow in CD vs HFD. They should also validate by PCR the modulation of ABCA1 mRNA expression. Also, how long such rapid metabolic adaptation of Lyve1+Tim4+ ATMs to HFD ingestion lasts? When is the system back to normal? Any synergy when fed multiple times with HFD? Also, upon Tim4 blockade and clodronate treatment, is expression of ABCA1 modulated? Gene expression profile of cluster 1 with or without Tim4 treatment would also be important to do to validate the proposed mechanism.

The authors did not really validate by flow cytometry the difference between cluster 2 vs cluster 3? What do they represent? Independent populations or different stages of activation/differentiation?

Minor:

The representation of figure 5B is could be confusing. On flow plots of aTim4 treated mice for example, there are 96% of Tim4⁻ cells but only 50% on the bar plot (as it is % of CD45). Authors should plot both quantifications as % of CD45 and % of CD45 F4/80^{high}

Please verify "While the raised post-prandial NEFA was independent of Tim4 and chloroquine, the rise in total cholesterol was not".

Reviewer #2:

Remarks to the Author:

In this manuscript by Magalhaes et al, the authors identify resident adipose tissue macrophages are important regulators of cholesterol efflux and post-prandial levels of plasma HDL cholesterol in mice. Using scRNAseq analysis of myeloid cells in chow-fed and high fat diet fed mice, the authors define multiple subsets of adipose tissue macrophages and identify a population of Lyve1+Tim4+ resident macrophages that highly express ABCA1. Antibody-mediated inhibition of Tim4 inhibited ATM lysosomal activity and the release of post-prandial HDLc following a high fat meal, implicating resident macrophages as important regulators of this process. Clodronate, which efficiently depletes hepatic and peritoneal macrophages, does not impact this process, leading the investigators to conclude that resident macrophages in the adipose are responsible. This approach lacks the specificity to make that conclusion. First, adipose tissue macrophages are reduced by 40% after clodronate treatment, which would be expected to have at least a partial effect if true. Second, it does not exclude a role for other tissue resident macrophage populations. Overall the findings are interesting, but not investigated in sufficient depth to justify the conclusions made.

General comments:

1. The authors have not cited prior work that is relevant to many aspects of the current manuscript. For example, the role of ABCA1 in adipose tissue has previously been investigated (PMIDs: 26531812, 29348118, 27420620, 30881070). Although much of this literature focuses on the role of ABCA1 in adipocytes, some of it implicates ATMs, and the idea that adipose tissue as an organ contributes to cholesterol efflux and plasma HDLc levels is not novel, and should be mentioned and properly referenced.

Furthermore, on page 3 the authors state that recruitment has been considered the main mechanism for the accumulation of macrophages in adipose tissue during prolonged HFD. This statement is incomplete, as many studies have established roles for both macrophage proliferation (PMID: 28108608, 27031964) and retention (PMID: 24584118) as processes that contribute significantly to macrophage accumulation in visceral adipose tissue. The contributions of these processes to macrophage accumulation should be included and cited.

Another example is the marked difference in the lipoprotein profiles of mice and humans. Mice carry the majority of their cholesterol in HDL particles, while in humans LDL particles are the main carriers in the circulation. This is not mentioned in the manuscript, but has major implications on the findings of the manuscript. The relevance of the findings in mice as they relate to human cholesterol metabolism are not discussed.

2. As described above, the data provided implicate a role for resident adipose tissue macrophages in regulating postprandial plasma levels of HDLc in mice, but fall short of convincingly demonstrating this. More specific approaches are needed, such as using adipose tissue transplantation from genetically engineered mouse lines (eg. Tim4-specific cre lines) or tissue targeted macrophage depletion strategies (diphtheria Toxin R mediated depletion directed to adipose tissue by injection strategies), and measurements of ABCA1-dependent cholesterol efflux in vitro and in vivo.

3. The authors report that Lyve1+Tim4+ resident macrophages do not express LXR - a key transcriptional regulator of ABCA1 expression in other tissue macrophages - nor do they express Lipa, a lysosomal lipase that plays a key role in the generation of free cholesterol for efflux in all other macrophages. These inconsistencies with the established literature on the regulation of ABCA1 and cholesterol efflux in other macrophages need to be addressed. As presented, the findings are not convincing, and further investigation of the mechanisms of ABCA1 regulation and efflux in Lyve1+Tim4+ macrophages are needed in light of these differences from other macrophage subsets.

Reviewer #3:

Remarks to the Author:

In the paper by Magalhaes et al., the authors describe, using single-cell RNA seq analysis, a novel macrophage subpopulation within the adipose tissue, which express high levels of F4/80, Lyve1 and TIM4. This population is shown to be a resident, self-renewing, and independent of monocyte contribution, similar to other macrophages in heart and other tissues. The authors investigated the role of this subpopulation in the context of post-prandial cholesterol trafficking by overnight high diet feeding. F4/80hiLyve1+TIM4+ are important for cholesterol metabolism, and their blockade by TIM4 antibody impaired cholesterol metabolism in a manner similar to chloroquine, a general lysosomal inhibitor.

Major critique:

The study is well performed and elegantly designed, followed by a rational and step-wise approach. The authors start by unbiased characterization ("atlas") of various adipose tissue macrophages, focusing on the F4/80hiLyve1+TIM4+ which is well characterized in terms of ontogeny and function, and the findings are of importance to the field of metabolism.

However, more mechanistic insight of how TIM4 is involved here, in the molecular/biochemical level is not clear. The authors show that TIM4 blockade by antibody, which is not depleting

F4/80hiLyve1+TIM4+ cells, can in itself impair cholesterol metabolism via reduction of lysosomal content. However, no mechanistic explanation is given as to what is the connection between Tim-4 and lysosomal activity of the TAM; is there an unknown TIM4 ligand, or a binding partner, that mediate this function? Would a system in which TIM4 is knocked-down or knocked-out will also show similar results? While the existence of a link between TIM4, lysosomal content and cholesterol metabolism is persuasive, how does TIM4 mediates this surprising and intriguing cellular pathway is not explored. Can RNAseq following perturbation (antibody blockade) provide a clue for the mechanism? I believe this would make the research much stronger.

Minor comments:

1. In the characterization of adipose tissue macrophages in Figure 1, can the authors describe the expression levels of Cx3cr1? Due to the existence of Cx3cr1 Cre-lox systems, it would be interesting to use them to nail down the ontogeny of adipose tissue macrophages. By this I do not mean that I expect the authors to use those tools, but it would be important for the community to know the expression levels of Cx3cr1. Similarly, Itgax (encoding for CD11c) should also be shown.
 2. In Figure 3C, it is very hard to judge from the histogram what are the difference in Lipidtox staining between the three macrophage subpopulations. I suggest removing these plots and use only the MFIs.
 3. The authors at one point look at macrophages in human subcutaneous fat. The visceral and subcutaneous fat are very distinct and different tissues and probably have also very different macrophage compartments. Throughout the mouse work in the paper, only visceral (epididymal) fat is used, therefore the inclusion of subcutaneous human fat is odd. I suggest removing it from the paper.
 4. Figure 5F is puzzling. The authors shown in Figure 5B that the TIM4 antibody blocks TIM4 and thus TIM4 cannot be detected by flow cytometry in F4/80hiTIM4+ cells. Therefore, how come the authors are able to look at Lysotracker in F4/80hiTIM4+ cells after anti-TIM4 treatment? Was this performed on total F4/80hi cells? Please clarify and perhaps also re-write this section of the Figures.
 5. Figure 6, although extensive, in not contributing much new data and is mostly showing negative data which, I feel is more technical. I suggest moving it to be a supplementary figure.
- In summary, the paper highlights an important function of Tim-4 in ATM in regulating lipid metabolism by affecting lysosomal function. How does Tim-4 achieve this function has not been addressed, which will provide an important insight in the role of Tim-4 and Tim-4+ ATM in dyslipidemia, lipid storage and adiposity.

Reviewer #1 (Remarks to the Author)

Magalhaes et al describe here the heterogeneity of adipose tissue macrophages in normal mice vs mice fed overnight with HFD to study the role of macrophages in regulating post-prandial cholesterol transport. They notably identify an adipose tissue resident macrophage population expressing Tim4 and ABCA1. The study is concise and of interest but however some important experiments or analysis are missing to provide a complete and convincing study.

1. The data presented in Figure 4 represent the most important part of the study (comparing changes in expression profile in normal mice vs mice fed overnight with HFD) but are not clearly presented. It is not clear to the reviewer if CD versus overnight HFD lead to significant and reproducible changes in cell composition as well as in gene expression profile across the defined clusters. It would be useful to display the data for HFD and compare them point by point / cell population by cell population to CD. As the point here is to show the adaptation of cluster 1 after HFD ingestion so the comparison between cluster 1 and 3 is useless (already done in the previous figures). Authors should insist on CD to HFD comparison here! and show differentially expressed genes etc...

We appreciate this comment and to increase clarity of the manuscript we now present the analysis of the effect of overnight HFD feeding on ATM clusters in a separate figure (Fig. 5) and have moved the analysis regarding the metabolic specificity of Lyve1⁺Tim4⁺ ATMs to Figure 3 (now panel 3E and 3F) and the results are now described earlier in the manuscript with the characterisation of Lyve1⁺Tim4⁺ ATMs (page 10).

We added to the manuscript a detailed paragraph describing the effect of overnight HFD on ATM cluster composition (page 11). Overnight HFD feeding led to an increase in the proportions of cluster 5, which represents 1.8% of all ATMs at CD and 7.8% after HFD and cluster 4, which represents 18% of all ATMs at CD and 26% after HFD (Fig. 1C). These clusters correspond to the most recent ATMs, which suggest that overnight HFD feeding increase recruitment of monocyte-derived ATMs. However, we found that overnight HFD feeding had no effect on the proportion of F4/80^{high}Tim4⁺, F4/80^{high}Tim4⁻ and F4/80^{low} ATM populations (Fig. S2E) and that increase in the proportion of monocyte-derived ATMs seen in scRNAseq was not reflected by an increase in the percentage of F4/80^{low} ATMs which encompass cluster 4, 5, 7 and 8.

The number of DEGs between mice fed overnight HFD and mice kept on CD in all clusters is shown in Fig. 5A. HFD feeding led to increase expression of a number of genes in ATMs from cluster 1 to 4 but had little impact on ATMs from cluster 5 to 8. Since our analysis revealed that most changes in pathway activation after HFD feeding were observed in cluster 1, we focused our analysis on this cluster and on genes involved in metabolic regulation. We now show 4 examples of DEGs between HFD and CD in cluster 1: *Abca1*, *Cd36*, *Hspa1a* and *Malat1* (Fig. 5C). Both *Hspa1a* and *Malat1* have been shown to regulate *Abca1* expression and could potentially be involved here (Liu, Molecular Medicine Reports 2020; Gungor, Mol Met 2019). We also show that in contrast to LAM (cluster 8), *Nr1h3* (LXRα) is not induced in Lyve1⁺Tim4⁺ ATMs (cluster 1). See manuscript page 12.

2. The authors show more ABCA1 mRNA in cluster 1 in CD vs HFD. But surprisingly they do not compare its expression by flow in CD vs HFD. They should also validate by PCR the modulation of ABCA1 mRNA expression.

We confirmed by qPCR increased expression of *Abca1* in cell-sorted F4/80^{high}Tim4⁺ ATMs from mice fed overnight with a HFD compared to mice kept on control diet (Fig. 6I). However, by flow-cytometry we found no difference in membrane ABCA1 expression (Fig. 6H). We also confirmed by flow-cytometric analysis that HFD feeding led to increase expression of CD36 (Fig. 6F) and Hsp70 (Fig. 6J), two of the factors which were highlighted in the scRNAseq analysis. See manuscript page 14 and 15.

3. Also, how long such rapid metabolic adaptation of Lyve1+Tim4+ ATMs to HFD ingestion lasts? When is the system back to normal? Any synergy when fed multiple times with HFD?

While these are very important questions, unfortunately they are beyond the scope of current study which aims to understand the role of ATMs and Tim4 in the metabolism of post-prandial lipids. Future studies will be required to address how obesity impacts the function of this population of ATMs, which we and others found to be maintained in obese mice (Fig. 2 and Jaitin et al, Cell 2019 and Moura Silva JEM 2019).

4. Also, upon Tim4 blockade and clodronate treatment, is expression of ABCA1 modulated? Gene expression profile of cluster 1 with or without Tim4 treatment would also be important to do to validate the proposed mechanism.

While we did not repeat the single-cell RNAseq experiment with Tim4 blockade, we validated some of the factors whose expression were changed by HFD feeding by flow-cytometry and RT-PCR and analysed the effect of Tim4 blockade and chloroquine treatment on ABCA1, CD36, HSP70. We also verified that *Nr1h3* was not changed. See manuscript page 14 and 15.

We found by RT-PCR that increase in *Abca1* expression in cell-sorted F4/80^{high}Tim4⁺ ATMs after overnight HFD feeding was dependent on Tim4 (Fig. 6I), suggesting that Tim4 activation/engagement regulates *Abca1* expression. We confirmed by RT-PCR that HFD feeding did not lead to induction of *Nr1h3* in F4/80^{high}Tim4⁺ ATMs and that its expression was independent of Tim4 (Fig. 6I). However, we found that membrane ABCA1 expression was not affected by Tim4 blockade. This may be due to the length of the experiment, too short to induce changes in protein expression.

We show that the increase in CD36 and HSP70 expression in F4/80^{high}Tim4⁺ ATMs after HFD feeding was independent on Tim4 but dependent on functional lysosomes. Interestingly, Tim4 blockade led to increased membrane expression of CD36 which suggested that Tim4 may be involved in the trafficking of CD36.

Taken together, these new results indicate that overnight HFD feeding leads to a rapid increase in CD36 expression in F4/80^{high}Tim4⁺ ATMs, expanding their capacity to uptake and process lipids. While Tim4 is not required to potentiate CD36 and HSP70 expression, it is critical to increase F4/80^{high}Tim4⁺ ATMs lysosomal function and *Abca1* expression after HFD feeding.

5. The authors did not really validate by flow cytometry the difference between cluster 2 vs cluster 3? What do they represent? Independent populations or different stages of activation/differentiation?

The level of expression of RELM α enables cells from cluster 2 and cluster 3 to be distinguished. See Figure 1E and 2B. A sentence has been added page 7 to clarify this point.

As stated in the manuscript, lineage inference with slingshot indicated that ATMs followed a pseudotime trajectory straddling cluster 5, 4, 3, 2 and 1 in mice kept on CD. Together with the BM chimeras showing that F4/80^{low} ATMs have a high turn-over and Tim4⁺ macrophages have long term residency (>8 weeks), these data, strongly support the hypothesis that these clusters represent different maturation stage of ATMs from ATM newly differentiated from monocytes (cluster 5) to mature ATM having spent a long time in adipose tissue (cluster 1). See manuscript page 5.

Minor:

The representation of figure 5B is could be confusing. On flow plots of aTim4 treated mice for example, there are 96% of Tim4⁻ cells but only 50% on the bar plot (as it is % of CD45). Authors should plot both quantifications as % of CD45 and % of CD45 F4/80^{high}.

Thank you for this comment, we acknowledge that figure 5B was unclear. Plots in what is now re-named as figure 6B represent the gating strategy used to calculate the percentage of the various ATM populations. To clarify this, an arrow has been added to show that Tim4⁺ and Tim4⁻ ATMs were gated from the F4/80^{high} ATM population, which were gated from CD45⁺ cells. The gate name (CD45⁺) has now been added.

Please verify “While the raised post-prandial NEFA was independent of Tim4 and chloroquine, the rise in total cholesterol was not”.

This sentence has now been clarified. See page 13.

Reviewer #2 (Remarks to the Author)

In this manuscript by Magalhaes et al, the authors identify resident adipose tissue macrophages are important regulators of cholesterol efflux and post-prandial levels of plasma HDL cholesterol in mice. Using scRNAseq analysis of myeloid cells in chow-fed and high fat diet fed mice, the authors define multiple subsets of adipose tissue macrophages and identify a population of Lyve1+Tim4+ resident macrophages that highly express ABCA1. Antibody-mediated inhibition of Tim4 inhibited ATM lysosomal activity and the release of post-prandial HDLc following a high fat meal, implicating resident macrophages as important regulators of this process. Chlodronate, which efficiently depletes hepatic and peritoneal macrophages, does not impact this process, *leading the investigators to conclude that resident macrophages in the adipose are responsible*. This approach lacks the specificity to make that conclusion. First, adipose tissue macrophages are reduced by 40% after chlodronate treatment, which would be expected to have at least a partial effect if true. Second, it does not exclude a role for other

tissue resident macrophage populations. Overall the findings are interesting, but not investigated in sufficient depth to justify the conclusions made.

While we agree that our results can not directly implicate ATMs in the regulation of post-prandial cholesterol levels, our results do show a role for Tim4 in this process and in the regulation of the lysosomal activity of Tim4⁺ ATMs. The title of the study reflects this. We have carefully screened the rest of manuscript for other over-interpretations and we have now amended the abstract to avoid any confusion:

“Thus, these data indicate that Tim4 is a key regulator of post-prandial cholesterol transport and ATM function and may represent a novel pathway to treat dyslipidemia.”

General comments:

1. The authors have not cited prior work that is relevant to many aspects of the current manuscript. For example, the role of ABCA1 in adipose tissue has previously been investigated (PMIDs: 26531812, 29348118, 27420620, 30881070). Although much of this literature focuses on the role of ABCA1 in adipocytes, some of it implicates ATMs, and the idea that adipose tissue as an organ contributes to cholesterol efflux and plasma HDLc levels is not novel, and should be mentioned and properly referenced.

We thank the reviewer for highlighting this work. The manuscript now introduces the role of ABCA1 in adipose tissue and in the development of diet-induced obesity. See page 3.

Furthermore, on page 3 the authors state that recruitment has been considered the main mechanism for the accumulation of macrophages in adipose tissue during prolonged HFD. This statement is incomplete, as many studies have established roles for both macrophage proliferation (PMID: 28108608, 27031964) and retention (PMID: 24584118) as processes that contribute significantly to macrophage accumulation in visceral adipose tissue. The contributions of these processes to macrophage accumulation should be included and cited.

We have now included the references showing that increased macrophage content in adipose tissue during obesity is dependent on increased recruitment, retention and in-situ proliferation. See page 3.

Another example is the marked difference in the lipoprotein profiles of mice and humans. Mice carry the majority of their cholesterol in HDL particles, while in humans LDL particles are the main carriers in the circulation. This is not mentioned in the manuscript, but has major implications on the findings of the manuscript. The relevance of the findings in mice as they relate to human cholesterol metabolism are not discussed.

This important difference between mice and humans is now mentioned in the discussion. See page 18-19.

2. As described above, the data provided implicate a role for resident adipose tissue macrophages in regulating postprandial plasma levels of HDLc in mice, but fall short of convincingly demonstrating this. More specific approaches are needed, such as using adipose tissue transplantation from genetically engineered mouse lines (eg. Tim4-specific cre lines) or tissue targeted macrophage depletion strategies (diphtheria Toxin R mediated depletion directed

to adipose tissue by injection strategies), and measurements of ABCA1-dependent cholesterol efflux *in vitro* and *in vivo*.

As noted above, we agree that we cannot firmly demonstrate that Tim4⁺ ATMs are responsible for the increase in postprandial HDLc levels. This problem will be difficult to address directly, since Tim4 is expressed by a subpopulation of resident macrophages in all tissues and there are no genetic tools allowing the specific targeting of Tim4⁺ ATMs. While a Tim4-Cre line exists and could be crossed with *Abca1*^{f/f} mice, this would only allow us to confirm that the effect on reverse cholesterol transport is mediated by ABCA1 and not confirm that the effect is mediated by ATMs, since Tim4⁺ macrophages are found in nearly all tissues. We thus think that the time and expense of generating such a murine model is beyond the scope of this study.

ATMs from all fat deposits are likely to be involved in post-prandial regulation of HDLc, which is a critical limitation to the fat depot grafting experimental approach.

We isolated and cultured *in vitro* Tim4⁺ ATMs to test cholesterol efflux. However, we found that ATMs quickly de-differentiated and lost expression of Tim4 when cultured in plates. This is a recurrent issue when working with tissue macrophages which limits the *in vitro* characterisation of tissue macrophage function. However, we have been able to demonstrate using freshly isolated ATMs in a 2 hour long *in vitro* culture experiment that CD36 was critical for the uptake of lipid particles (here we used BODIPY-LDL) but not Tim4. Uptake of BODIPY-LDL was associated with an increase in lysotracker MFI, which was dependent on Tim4 (Fig. S5A). This mirrors our *in vivo* experiments. In tumor-associated macrophages, Tim4 was also shown to be dispensable for the uptake of apoptotic tumor cells but to be critical for lysosomal activation and the degradation of ingested tumor cells (Baghdadi, Immunity 2013). See results page 14 and discussion page 18.

We confirmed by flow-cytometry using Annexin V staining that the lipoprotein particles LDL and chylomicrons are covered on their surface with the phospholipid phosphatidylserine (PS) (Fig. S5C). While the interaction of Tim4 with PS on lipoprotein particles may not be involved in their cellular uptake it may mediate their trafficking to the lysosomes. See manuscript page 14.

These new data thus provide new insights into the mechanisms governing Tim4 mediated regulation of postprandial cholesterol and highlight the importance of CD36 for lipoprotein uptake and of Tim4 for the downstream handling of lipoprotein particles by lysosomes. See discussion page 17.

3. The authors report that Lyve1+Tim4+ resident macrophages do not express LXR - a key transcriptional regulator of ABCA1 expression in other tissue macrophages – nor do they express Lipa, a lysosomal lipase that plays a key role in the generation of free cholesterol for efflux in all other macrophages. These inconsistencies with the established literature on the regulation of ABCA1 and cholesterol efflux in other macrophages need to be addressed. As presented, the findings are not convincing, and further investigation of the mechanisms of ABCA1 regulation and efflux in Lyve1+Tim4+ macrophages are needed in light of these differences from other macrophage subsets.

We were also puzzled to find that Tim4⁺ ATMs which express ABCA1 did not express *Nr1h3* or *Lipa*. The dissociation of the expression of *Abca1* and *Nr1h3* in Lyve1+Tim4⁺ ATMs is

confirmed by the Immgen database which showed that in contrast to liver macrophages which highly express both *Abca1* and *Nr1h3*, ATMs expressed high levels of *Abca1* and no *Nr1h3* (Fig. S4). See manuscript page 10.

We also confirmed by qPCR (Fig. 6I) that the levels of expression of *Nr1h3* is extremely low, and is not up-regulated by HFD. It is thus unlikely that NR1H3 regulates *Abca1* expression. See manuscript page 15.

We also compared the level of expression of ABCA1 in Tim4⁺ ATMs with Tim4⁺ Kupffer cells of the liver and Tim4⁺ peritoneal cavity macrophages and found that Tim4⁺ ATMs express twice as much ABCA1 than these two Tim4⁺ macrophage populations. In the peritoneal cavity, Tim4⁻ macrophages showed limited ABCA1 expression compared to Tim4⁺ macrophages which support the hypothesis that Tim4 is linked to ABCA1 expression in resident macrophages (Fig. S5D). High ABCA1 expression on Tim4⁺ ATM is thus characteristic of this population of resident macrophages and may be regulated by Tim4 as our Tim4 blockade experiment in mice fed overnight HFD suggests. See manuscript page 16.

We now provide additional immunofluorescence staining showing in wholemount adipose tissue that ABCA1 was concentrated in some areas of the cytoplasmic membrane or intracellular membranes which were in contact with lysosomes and Tim4 (Fig. 4C). This suggests that in Tim4⁺ ATMs, a mechanism enables excess cholesterol from cholesterol rich lipoprotein particles to be transferred from lysosomes to the cytoplasmic membrane and ABCA1 for export. See results page 11 and discussion page 18.

We now show that Tim4⁺ ATMs express 2 other factors which have been implicated in the regulation of *Abca1* expression: (i) The long coding RNA Malat1 which a report linked to increased expression of ABCA1 through modulation of microRNA-17-5p (Liu, Molecular Medicine Reports 2020) and (ii) HSP70 which has been shown to increase *Abca1* expression and cholesterol efflux in cholesterol-laden macrophages (Gungor, Mol Met 2019). In the study by Gungor *et al*, HSP70 led to a strong increase in *Abca1* and ABCA1 expression, while only marginally affecting *Nr1h3* expression. Future studies will explore how these factors regulate *Abca1* expression in ATMs. See Fig. 5C and 6J and manuscript page 12 and 15.

Reviewer #3 (Remarks to the Author):

In the paper by Magalhaes et al., the authors describe, using single-cell RNA seq analysis, a novel macrophage subpopulation within the adipose tissue, which express high levels of F4/80, Lyve1 and TIM4. This population is shown to be a resident, self-renewing, and independent of monocyte contribution, similar to other macrophages in heart and other tissues. The authors investigated the role of this subpopulation in the context of post-prandial cholesterol trafficking by overnight high diet feeding. F4/80hiLyve1+TIM4+ are important for cholesterol metabolism, and their blockade by TIM4 antibody impaired cholesterol metabolism in a manner similar to chloroquine, a general lysosomal inhibitor.

Major critique:

The study is well performed and elegantly designed, followed by a rational and step-wise approach. The authors start by unbiased characterization (“atlas”) of various adipose tissue macrophages, focusing on the F4/80hiLyve1+TIM4+ which is well characterized in terms of ontogeny and function, and the findings are of importance to the field of metabolism.

However, more mechanistic insight of how TIM4 is involved here, in the molecular/biochemical level is not clear. The authors show that TIM4 blockade by antibody, which is not depleting F4/80hiLyve1+TIM4+ cells, can in itself impair cholesterol metabolism via reduction of lysosomal content. However, no mechanistic explanation is given as to what is the connection between Tim-4 and lysosomal activity of the TAM; is there an unknown TIM4 ligand, or a binding partner, that mediate this function? Would a system in which TIM4 is knocked-down or knocked-out will also show similar results? While the existence of a link between TIM4, lysosomal content and cholesterol metabolism is persuasive, how does TIM4 mediates this surprising and intriguing cellular pathway is not explored. Can RNAseq following perturbation (antibody blockade) provide a clue for the mechanism? I believe this would make the research much stronger.

While we did not repeat the single-cell RNAseq experiment by including Tim4 blockade, we validated some of the factors whose expression were changed by HFD feeding by flow-cytometry and RT-PCR and analysed the effect of Tim4 blockade on CD36, HSP70, ABCA1 and *Nr1h3*. In particular, we now show that the potentiation of *Abca1* expression by HFD feeding is dependent on Tim4 and not on NR1H3. See Reviewer 1 point 4.

We now provide additional data regarding the contribution of Tim4 to the processing of cholesterol-rich lipoprotein particles and the role of CD36 in their uptake which are detailed in answer to reviewer 2 point 2 and summarised here:

- CD36 is required for *in vitro* LDL uptake by Tim4⁺ ATMs. Tim4 was not required for LDL uptake but was required to increase lysosomal content following LDL uptake. This complements *in vivo* data showing increased lysosomal content following HFD feeding dependent on Tim4.
- HFD feeding leads to increase in CD36 expression and this is not dependent on Tim4. Tim4 blockade leads to increased membrane expression of CD36 which suggests that Tim4 may be involved in the trafficking of CD36.
- The lipoprotein particles LDL and chylomicrons are covered on their surface with phosphatidylserine, the ligand of Tim4.

These new data thus provide new insights into the mechanisms governing Tim4 mediated regulation of postprandial cholesterol and highlight the importance of CD36 for lipoprotein uptake and of Tim4 for the downstream handling of lipoprotein particles by lysosomes.

Minor comments:

1. In the characterization of adipose tissue macrophages in Figure 1, can the authors describe the expression levels of Cx3cr1? Due to the existence of Cx3cr1 Cre-lox systems, it would be interesting to use them to nail down the ontogeny of adipose tissue macrophages. By this I do not mean that I expect the authors to use those tools, but it would be important for the community to know the expression levels of Cx3cr1. Similarly, Itgax (encoding for CD11c) should also be shown.

Itgax expression is shown in Fig. S1A. Its expression was low on all ATM subsets. *Cx3cr1* expression is limited to cluster 6 which also express high levels of *Ccr2* and *Plac8* and is now shown in Fig. S1A. See manuscript page 6.

2. In Figure 3C, it is very hard to judge from the histogram what are the difference in Lipidtoxin staining between the three macrophage subpopulations. I suggest removing these plots and use only the MFIs.

We think it is important to show representative staining for all cytometry experiments, so that the reader can appreciate the level of staining for the molecule of interest.

3. The authors at one point look at macrophages in human subcutaneous fat. The visceral and subcutaneous fat are very distinct and different tissues and probably have also very different macrophage compartments. Throughout the mouse work in the paper, only visceral (epididymal) fat is used, therefore the inclusion of subcutaneous human fat is odd. I suggest removing it from the paper.

We agree that subcutaneous and visceral adipose tissue are very different type of tissues. However, we think that the fact that Tim4^+ ATMs can be found in both type of adipose tissues is important and may be an indication that Tim4-dependent reverse cholesterol transport may occur in both tissues.

4. Figure 5F is puzzling. The authors shown in Figure 5B that the TIM4 antibody blocks TIM4 and thus TIM4 cannot be detected by flow cytometry in F4/80 $^{\text{hi}}$ TIM4 $^+$ cells. Therefore, how come the authors are able to look at LysoTracker in F4/80 $^{\text{hi}}$ TIM4 $^+$ cells after anti-TIM4 treatment? Was this performed on total F4/80 $^{\text{hi}}$ cells? Please clarify and perhaps also re-write this section of the Figures.

As mentioned in the figure legend, to allow detection of Tim4 in mice receiving anti-Tim4 Ig, Tim4 was detected using rat anti-Tim4 Ig plus secondary anti-Rat Ig-647 and Miltenyi's humanised antibodies were used to stain for F4/80, CD11b and CD45. We have now added a sentence in the results page 13.

5. Figure 6, although extensive, is not contributing much new data and is mostly showing negative data which, I feel is more technical. I suggest moving it to be a supplementary figure.

While presenting negative data, this figure is important since it allows us to exclude a role for Tim4^+ macrophages of the liver and peritoneal cavity in the regulation of post-prandial cholesterol. For this reason we think it is important to keep it as a main figure.

In summary, the paper highlights an important function of Tim-4 in ATM in regulating lipid metabolism by affecting lysosomal function. How does Tim-4 achieve this function has not been addressed, which will provide an important insight in the role of Tim-4 and Tim-4^+ ATM in dyslipidemia, lipid storage and adiposity.

Reviewers' Comments:

Reviewer #1:

Remarks to the Author:

The authors have addressed the issues that were raised.

Reviewer #2:

Remarks to the Author:

The authors have responded to the reviewers comments with some new data, but for the most part have addressed the critiques with written responses. They have tempered their conclusions about ABCA1-mediated cholesterol efflux throughout the manuscript and the manuscript title should also be adjusted accordingly. My original concerns about the limited mechanistic insight of their findings still stand, but the manuscript serves as a thorough description of the macrophage subsets in the adipose tissue and their potential roles in post-prandial lipid metabolism.

Reviewer #3:

Remarks to the Author:

I went through the revised paper and I think I have no further comments, the authors provided mechanistic data as we requested in the original review.

We would like to thank all three reviewers for their insightful comments, which helped us improve our manuscript.

Reviewer #1 (Remarks to the Author):

The authors have addressed the issues that were raised.

Reviewer #2 (Remarks to the Author):

The authors have responded to the reviewers comments with some new data, but for the most part have addressed the critiques with written responses. They have tempered their conclusions about ABCA1-mediated cholesterol efflux throughout the manuscript and the manuscript title should also be adjusted accordingly. My original concerns about the limited mechanistic insight of their findings still stand, but the manuscript serves as a thorough description of the macrophage subsets in the adipose tissue and their potential roles in post-prandial lipid metabolism.

We have now modified our title: “Role of Tim4 in the regulation of ABCA1⁺ adipose tissue macrophages and post-prandial cholesterol levels”.

Reviewer #3 (Remarks to the Author):

I went through the revised paper and I think I have no further comments, the authors provided mechanistic data as we requested in the original review.